# GeoCon: Compositional Generalization Through Geometric Constraints on Representation Structure

## Abstract

Compositional generalization, referring to the capacity to generalize novel combinations of fundamental and essential concepts, is thought to be the mechanism underlying a human's remarkable ability of rapid generalization to new knowledge and tasks. Recent research on brain neural activation space has found that the geometric structure of neural representations is highly related to human compositional generalization capability. In this paper, we extend the above observations from neuroscience to deep neural networks to validate the potential relationship between the geometric structure of representations and compositional generalization capability. In particular, we first construct a new compositional generalization benchmark from the existent datasets, which aims to discriminate multiple concepts simultaneously through a powerful representation. Meanwhile, for the aforementioned geometric constraint, the parallelism score is formally defined for deep neural networks. Subsequently, we decompose the deep neural network into two parts: the featurizer and the classifier, to investigate the relationship between compositional generalization capability and parallelism score separately. Our proposed method, **Geo**metric **Con**straint (GeoCon), involves distance variance minimization on the classifier and parallelism score maximization on the featurizer. Experiments on synthetic and real-world datasets demonstrate significant improvement of our approach, verifying the effectiveness of our neuroscience-inspired GeoCon approach towards human-like superior generalization ability.

## 1 Introduction

Humans exhibit a remarkable capacity for generalization by transferring existing limited prior knowledge to novel contexts. One underlying mechanism is hypothesized to be compositional generalization (Cole et al., 2013; Frankland & Greene, 2020; Hupkes et al., 2020), the ability to systematically disentangle learned concepts and recombine them into unseen compositions (e.g., *red apple* and *yellow banana* can be decoupled and recomposed into a new composition *yellow apple*). This ability, described by Chomsky (2014) as "the infinite use of finite means", is considered an essential characteristic of human intelligence. Despite the substantial advancements accomplished by deep neural networks (Li et al., 2021; Han et al., 2022), they still struggle with generalization performance and have been criticized for lacking compositional generalization capability, even when provided with extensive training data (Zhang et al., 2023). Consequently, it is a significant yet challenging research topic to study the compositional generalization mechanism of deep neural networks (Lin et al., 2023), which is crucial for advancing toward artificial intelligence.

**Related Work.** In previous research on compositional generalization in computer vision, one important related research field is disentangled representation learning (Higgins et al., 2017; Wang et al., 2022), which aims to extract independent underlying concept factors from mixed representations and recombine them to generate novel concept compositions predominantly on synthetic datasets. However, it remains unclear whether disentanglement can assist in compositional generalization, while some studies suggest that there is currently no evidence that explicitly decoupling input compositional factors substantially improves the learning efficiency or generalization capacity of models (Montero et al., 2020; Schott et al., 2021; Xu et al., 2022), whereas some claim to find a correlation between disentanglement and compositional generalization (Higgins et al., 2017; Esmaeili et al.,

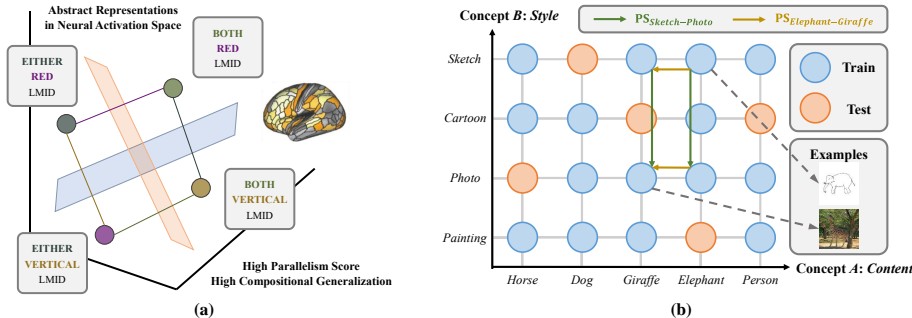

Figure 1: (a). Hypothetical geometric configurations of neural activation space. Parallel abstract representation leads to high compositional generalization capability. (b). Our designed compositional generalization benchmark. The model is expected to simultaneously differentiate two concepts.

2019; Mahon et al., 2023). Compositional zero-shot learning attempts to address the compositionality challenge in real-world scenarios, enhancing the accuracy of unseen attribute-object pairs when trained on different attribute-object pairs (Mancini et al., 2021; Wang et al., 2023; Hao et al., 2023). The objective of this task is to acquire invariant representations of objects while eliminating spurious attribute features, rather than systematically recognizing each concept, including attributes and objects. With the advancement of vision-language models (Zhang et al., 2024), several efforts have evaluated the compositional generalization capability of multimodal models (Ma et al., 2023; Yuksekgonul et al., 2023) and endeavored to develop novel training paradigms to enhance this ability (Zheng et al., 2024; Mitra et al., 2024). However, these purported improvements have been revealed to stem from linguistic priors rather than genuine enhancements in visual compositional generalization capability (Wu et al., 2023). Furthermore, extant visual encoders still perform inadequately in capturing compositional details (Tong et al., 2024). In summary, the internal mechanism of compositional generalization in visual models remains elusive.

To demystify the mechanism of compositional generalization from a neuroscience perspective, Bernardi et al. (2020) investigates the neural activation space in the hippocampus and prefrontal cortex, proposing the compositional additive representation extracted from neural signals, identified as the abstract representation. They further propose the parallelism score to quantify the parallelism of the geometric structure for the abstract representation, which is positively related to the compositional generalization capability, as shown in Fig.1.(a). The subsequent study (Ito et al., 2022) leverages the parallelism score to measure the fMRI activity signals of the human brain during the execution of tasks that necessitate logical decision, semantic comprehension, and motor response.

**Our Contributions.** Motivated by the aforementioned observations from neuroscience research, we would like to validate whether this pattern is consistent in deep neural networks. Initially, we establish a novel compositional generalization task from the existent datasets and provide a formal definition of the parallelism score. Afterward, we partition the deep neural network into the featurizer and the classifier, proposing regularization techniques: distance variance minimization on the classifier and parallelism score maximization on the featurizer, to constrain the representation space. As a result, we introduce the **Geo**metric **Con**straint (GeoCon) to strengthen the visual compositional generalization capability. Experimental results demonstrate that our GeoCon method surpasses the current baselines across multiple datasets. This research endeavor has the potential to serve as a valuable investigation into the mechanisms underlying compositional generalization, thereby advancing the development of deep neural networks toward achieving human-like intelligence.

## 2 PRELIMINARIES

### 2.1 COMPOSITIONAL GENERALIZATION

To achieve compositional generalization, it is imperative to systematically differentiate each concept and preserve the discrimination capacity for novel combinations. To facilitate comprehension and maintain simplicity, we shall initially examine the case of two concept factors, $A$ and $B$. This framework can subsequently be expanded to encompass additional concepts.

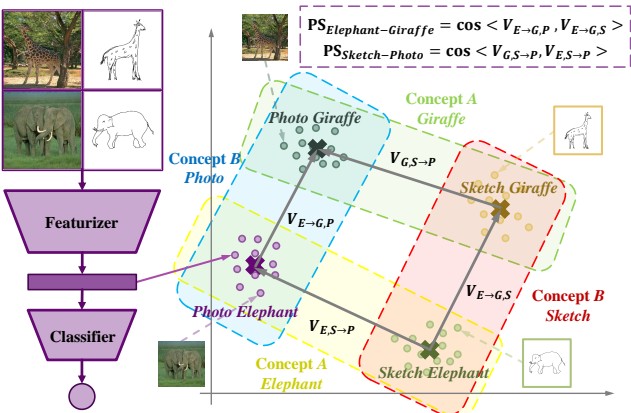

Figure 2: Illustration for the parallelism score calculation. We first calculate the expected representation for each combination, then obtain the concept transform vectors within diverse conceptual contexts (e.g., $\boldsymbol{V}_{E \to G,P}$ represents the concept transform vector that changes one concept from *Elephant* to *Giraffe* in the setting of *Photo* for another concept). Finally, we measure the parallelism of these concept transform vectors using the cosine similarity.

Consider predicting the target $\boldsymbol{a} = (a_1, a_2, \ldots, a_{m_A}) \in \mathbb{R}^{m_A}$ with $m_A$ concepts and target $\boldsymbol{b} = (b_1, b_2, \ldots, b_{m_B}) \in \mathbb{R}^{m_B}$ with $m_B$ concepts simultaneously from input image $\boldsymbol{x} \in \mathbb{R}^{L \times H \times C}$. The featurizer can be defined as $g : \mathbb{R}^{L \times H \times C} \mapsto \mathbb{R}^d$ that maps an input image $\boldsymbol{x}$ into the $d$-dimensional representation space, while classifiers can be characterized as $f_A : \mathbb{R}^d \mapsto \mathbb{R}^{m_A}$ and $f_B : \mathbb{R}^d \mapsto \mathbb{R}^{m_B}$ that map a representation $\boldsymbol{r} = g(\boldsymbol{x})$ into $A$ and $B$ concept space respectively. The objective of this task is to develop the model $f_A \circ g$ and $f_B \circ g$ capable of accurately predicting multiple target concepts simultaneously, utilizing representations extracted by a single featurizer. In particular, to verify that $g$ captures a good representation, we consider $f_A$ and $f_B$ to be linear functions to eschew further modifications of the representation structure.

Let $\mathcal{X}$ denote a nonempty input image space, as well as $\mathcal{A}$ and $\mathcal{B}$ two target spaces of concepts $A$ and $B$. In the training process, we have training points $\mathcal{D}_{tr} = \{(\boldsymbol{x}^{(i)}, \boldsymbol{a}^{(i)}, \boldsymbol{b}^{(i)})\}_{i=1}^n$ sampled from distribution $\mathcal{D} = \mathcal{X} \times \mathcal{A} \times \mathcal{B}$. In the in-distribution generalization, the train set targets encompass all compositions within the concept space. However, referring to the definition of compositional generalization in Mahon et al. (2023); Xu et al. (2022), as demonstrated in Fig.1.(b), the train set targets will encounter each potential concept individually (containing all *styles* and *contents*), but there will still exist unseen concept combinations, which can be formally elucidated as:

$$\exists i \in \{1, 2, \ldots, m_A\}, j \in \{1, 2, \ldots, m_B\} \quad \text{s.t. } a_i \in \mathcal{A}_{tr}, b_j \in \mathcal{B}_{tr}, (a_i, b_j) \notin (\mathcal{A}_{tr} \times \mathcal{B}_{tr}) \quad (1)$$

where $\mathcal{A}_{tr}$ and $\mathcal{B}_{tr}$ denotes the training target spaces. The leftover space $(\mathcal{A}_{te} \times \mathcal{B}_{te}) = (\mathcal{A} \times \mathcal{B}) \setminus (\mathcal{A}_{tr} \times \mathcal{B}_{tr})$ is identified as the out-of-combination (OOC) in the target space. An ideal model exhibiting high compositional generalization capability can effectively learn from samples that only cover partial concept compositions but perform proficiency in OOC cases $\mathcal{D}_{te} = \mathcal{X}_{te} \times \mathcal{A}_{te} \times \mathcal{B}_{te}$, which characterize the test set, $\mathcal{X}_{te}$ denoting the corresponding input image space for $\mathcal{A}_{te} \times \mathcal{B}_{te}$.

Let $\mathcal{L}_{\mathcal{A}}(f_A, g; \mathcal{D}_{tr})$ and $\mathcal{L}_{\mathcal{B}}(f_B, g; \mathcal{D}_{tr})$ be cross-entropy loss functions that measures the discrepancy between the predictions and the targets, respectively (keeping consistent in Section 3). Accordingly, the compositional generalization can be formulated as the following optimization problem:

$$\min_{f_A, f_B, g} \mathbb{E}_{(\boldsymbol{x}, \boldsymbol{a}, \boldsymbol{b}) \in \mathcal{D}_{tr}} \Big[ \mathcal{L}_{\mathcal{A}}\big(f_A \circ g(\boldsymbol{x}), \boldsymbol{a}\big) + \mathcal{L}_{\mathcal{B}}\big(f_B \circ g(\boldsymbol{x}), \boldsymbol{b}\big) \Big] \quad (2)$$

## 2.2 PARALLELISM SCORE

We here provide a mathematical description for the parallelism score (PS) proposed qualitatively in neuroscience by Bernardi et al. (2020). Assign $\mathcal{X}(a, b) = \{\boldsymbol{x} \mid (\boldsymbol{x}, \boldsymbol{a}, \boldsymbol{b}) \in \mathcal{D}, (\boldsymbol{a}, \boldsymbol{b}) = (a, b)\}$ as the input samples with the target $(a, b)$. Subsequently, we obtain the expected representation indicating the centroid for points with the target $(a, b)$ as:

$$\overline{\boldsymbol{r}}(a, b) = \mathbb{E}_{\boldsymbol{x} \in \mathcal{X}(a, b)}[g(\boldsymbol{x})] \quad (3)$$

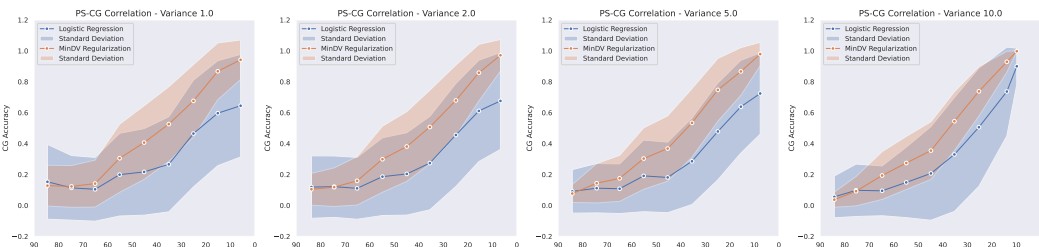

Figure 3: Positive correlation between PS and CG for logistic regression and MinDV regularization across the different variances for the Gaussian distributions. Since PS is nonlinear with respect to angle, the horizontal axis is converted to linear scale by the $\arccos$ function for clearer visualization.

Next, we introduce the concept transform vector of $a \rightarrow a'$ under context $b$, which measures the directional transformation of the expected representation of $(a, b)$ to that of $(a', b)$ as:

$$\boldsymbol{V}_{a \rightarrow a', b} = \boldsymbol{V}(a \rightarrow a'|b) = \overline{\boldsymbol{r}}(a', b) - \overline{\boldsymbol{r}}(a, b) \tag{4}$$

Correspondingly, the concept transform vector of $b \rightarrow b'$ under context $a$ can be described as:

$$\boldsymbol{V}_{a, b \rightarrow b'} = \boldsymbol{V}(b \rightarrow b'|a) = \overline{\boldsymbol{r}}(a, b') - \overline{\boldsymbol{r}}(a, b) \tag{5}$$

The parallelism score (PS) intents to quantify the consistency of concept transform vector directions across diverse contexts, utilizing the cosine similarity function as a measurement, as illustrated in Fig.2. Consequently, it is feasible to define the specific-concept-level $\text{PS}_{a \rightarrow a'}$, the overall-concept-level $\text{PS}_A$, and the dataset-level $\text{PS}_{\mathcal{D}}$ as follows:

$$\text{PS}_{a \rightarrow a'} = \text{PS}(a \rightarrow a') = \frac{1}{M_B} \sum_{b \neq b' \in \mathcal{B}} \cos \left\langle \boldsymbol{V}(a \rightarrow a'|b), \boldsymbol{V}(a \rightarrow a'|b') \right\rangle \tag{6}$$

$$\text{PS}_A = \text{PS}(\boldsymbol{a}) = \frac{1}{M_A} \sum_{a \neq a' \in \mathcal{A}} \text{PS}(a \rightarrow a') \tag{7}$$

$$\text{PS}_{\mathcal{D}} = \frac{M_A}{M_A + M_B} \text{PS}(\boldsymbol{a}) + \frac{M_B}{M_A + M_B} \text{PS}(\boldsymbol{b}) \tag{8}$$

where $M_A$ is the quantity of pairs for $a \neq a'$ and $M_B$ is the quantity of pairs for $b \neq b'$. The PS approaching 1 indicates highly parallel concept transform vectors, which may potentially lead to improved OOC performance, as suggested by Bernardi et al. (2020). This characteristic exhibits the potential to facilitate the training of models with enhanced compositional generalization capability.

## 3 METHODOLOGY

In this section, we aim to validate the effectiveness of the parallelism score (PS) from neuroscience on deep neural networks by investigating the correlation between PS and compositional generalization (CG) capability. Specifically, we partition the neural network into a featurizer that extracts representations and a classifier that makes final decisions based on these representations linearly. We seek to address the following inquiries: What kind of classifier can enhance robust CG capability? What kind of featurizer can yield high PS representations?

### 3.1 CLASSIFIER: DISTANCE VARIANCE MINIMIZATION

**Simulation Studies on Synthetic Datasets.** Assuming the existence of representations captured by the featurizer, we aim to explore the relationship between representation geometric structure and CG capability. Considering the simplest scenario involving two targets $\boldsymbol{a} = (a_1, a_2)$ and $\boldsymbol{b} = (b_1, b_2)$ with two-dimensional representations that can be visualized on a plane, on which we sample four points on a unit circle and consider them as centroids of representations with the concept combinations $(a_1, b_1), (a_1, b_2), (a_2, b_1),$ and $(a_2, b_2)$. By controlling the sampled points, we can manipulate the PS of the representation centroids. We further sample points around these centroids from the

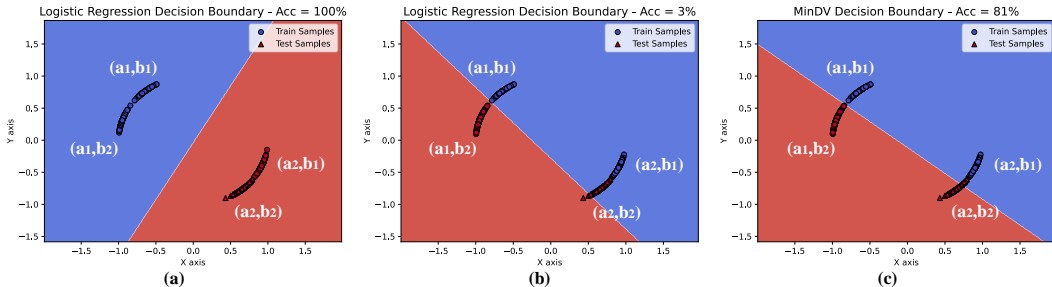

Figure 4: A fail case occurs when the PS of the sample points is 0.99. (a). Logistic regression exhibits 100% accuracy when classifying concepts $a_1$ and $a_2$. (b). Logistic regression has only 3% accuracy when classifying concepts $b_1$ and $b_2$. (c). After applying MinDV regularization, the accuracy improves to 81% when classifying concepts $b_1$ and $b_2$.

Gaussian distributions along the proximal circumference, to facilitate the generation of a synthetic dataset of representations with different PS, all retaining the same concept labels as the centroids.

Subsequently, the points centered at $(a_2, b_2)$ are designated as the test set, while points centered at $(a_1, b_1)$, $(a_1, b_2)$ and $(a_2, b_1)$ constitute the train set. We train two linear classifiers to classify $\boldsymbol{a} = (a_1, a_2)$ and $\boldsymbol{b} = (b_1, b_2)$. During the training process, the model is exposed to all concepts $a_1$, $a_2$, and $b_1$, $b_2$. Nevertheless, it has not encountered the combination of $(a_2, b_2)$, which is defined as the out-of-combination for CG. The mean and standard deviation of the accuracy of simultaneously classifying the targets correctly within different PS intervals are recorded, as shown in Fig.3.

The blue line in Fig.3 illustrates the relationship between PS and CG when logistic regression is employed as the linear classifier. As the PS increases, the model's CG accuracy also improves, demonstrating a positive correlation. A higher PS suggests that the configuration of the centroids more closely approximates a parallelogram rather than an irregular quadrilateral, resulting in greater parallelism between edges and making it more feasible for the model to accomplish CG tasks. The blue shaded area represents the standard deviation in accuracy, a larger standard deviation implies instability in CG accuracy. This observation suggests that even when the representations exhibit high PS, the model retains a considerable probability of demonstrating poor CG capability.

We conduct a detailed analysis of cases where CG fails under high PS representations. In Fig.4.(a), when classifying concepts $a_1$ and $a_2$, the distance between them is significant, allowing for distinct classification. However, a severe failure occurs when classifying concepts $b_1$ and $b_2$ with an accuracy of only 3%, as depicted in Fig.4.(b). During the training process, the logistic regression model tends to maximize the aggregate distances from all train sample points (circle points) to the decision boundary. Nevertheless, while classifying $(a_2, b_1)$ and $(a_2, b_2)$, since the test sample points (triangle points) centered around $(a_2, b_2)$ are unseen, the model will attempt to make the decision boundary as far away as possible from the visible $(a_2, b_1)$-centered points (meanwhile getting closer to the unseen $(a_2, b_2)$-centered points), in order to achieve enhanced basic generalization (albeit at the expense of the CG capability). To address the aforementioned challenges, we propose distance variance minimization regularization on the classifier to enhance the CG capability.

**Distance Variance Minimization.** The vanilla classifier demonstrates a failure on out-of-combination samples, due to neglecting the highly parallel geometric structures of representations. In this case, we aim for all samples to have as similar distances to the decision boundary as possible, essentially minimizing the variance of distances. This strategy can prevent the model from deviating excessively from the $(a_2, b_2)$-center points, based on which we propose **D**istance **V**ariance **Min**imization (MinDV) regularization to achieve this insight.

In more detail, when $f$ is a linear function, defined as $f(\boldsymbol{x}) = \boldsymbol{w}^T \boldsymbol{r} + \boldsymbol{b}$, where $\boldsymbol{r} = g(\boldsymbol{x})$, we know that the distance between the point and the decision boundary is $|\boldsymbol{w}^T \boldsymbol{r} + \boldsymbol{b}|/\|\boldsymbol{w}\|_2$. The distance variance (DV) of the samples from the decision boundary on dataset $\mathcal{D}$ can be denoted as:

$$\mathrm{DV}(f, g; \mathcal{D}) = \mathrm{Var}_{\boldsymbol{x} \in \mathcal{X}}\left[\frac{|\boldsymbol{w}^T g(\boldsymbol{x}) + \boldsymbol{b}|}{\|\boldsymbol{w}\|_2}\right] \tag{9}$$

Smaller DV indicates that the decision boundary is more parallel to the concept transform vectors.

Ideally, we aim to solve the following constrained optimization problem. If this objective is feasible, then our solution can rectify the failures. We generalize this finding in the subsequent theorem.

$$\min_{f_A,f_B,g} \mathbb{E}_{(\boldsymbol{x},\boldsymbol{a},\boldsymbol{b})\in\mathcal{D}_{tr}}\left[\mathcal{L}_{\mathcal{A}}(f_A, g; \mathcal{D}_{tr}) + \mathcal{L}_{\mathcal{B}}(f_B, g; \mathcal{D}_{tr})\right]$$
$$\text{s.t.}\quad \text{DV}(f_A, g; \mathcal{D}_{tr}) = 0,\ \text{DV}(f_B, g; \mathcal{D}_{tr}) = 0 \tag{10}$$

**Theorem 1.** *If we do not consider the stochastic noise of the feature distribution, assume that $PS = 1$, and the representation space is linearly separable, then $f$ calculated by Eq.10 is guaranteed to have 100% test accuracy, while vanilla classifier may fail for some cases.*

Provided that $\text{DV} = 0$ is extremely challenging in practice, we relax the formulation constraint to:

$$\min_{f_A,f_B,g} \mathbb{E}_{(\boldsymbol{x},\boldsymbol{a},\boldsymbol{b})\in\mathcal{D}_{tr}}\left[\mathcal{L}_{\mathcal{A}}(f_A, g; \mathcal{D}_{tr}) + \mathcal{L}_{\mathcal{B}}(f_B, g; \mathcal{D}_{tr})\right]$$
$$\text{s.t.}\quad \text{DV}(f_A, g; \mathcal{D}_{tr}) \leq \epsilon,\ \text{DV}(f_B, g; \mathcal{D}_{tr}) \leq \epsilon \tag{11}$$

where $\epsilon$ is the tolerance coefficient. The above is equivalent to the Lagrange function with appropriate hyperparameters $\alpha_A$ and $\alpha_B$, where the last terms are identified as the MinDV regularization:

$$\min_{f_A,f_B,g} \mathbb{E}\left[\mathcal{L}_{\mathcal{A}}(f_A, g; \mathcal{D}_{tr}) + \mathcal{L}_{\mathcal{B}}(f_B, g; \mathcal{D}_{tr})\right] + \alpha_A \text{DV}(f_A, g; \mathcal{D}_{tr}) + \alpha_B \text{DV}(f_B, g; \mathcal{D}_{tr}) \tag{12}$$

The MinDV regularization promotes model consistency in the distance from the decision boundary across all samples. The idea of "balance enhances generalization" has also been applied in domain generalization. For instance, the Invariant Risk Minimization (IRM) (Arjovsky et al., 2019) algorithm aims to achieve similar classification performance across diverse domains while maintaining overall classification efficacy. In summary, both MinDV and IRM are based on the assumption that there exists a substantial disparity between the observed training distribution and the unseen testing distribution. They strive to ensure that the model performs similarly across all visible distributions, rather than significantly outperforming on a specific distribution.

After utilizing MinDV regularization, the previous failed case is rectified effectively, with performance improving from 3% to 81%, as shown in Fig.4.(c). Moreover, the orange line in Fig.3 depicts the results after incorporating MinDV regularization. Concomitant with the increase in CG accuracy, the standard deviation is effectively controlled, indicating a reduction in the quantity of failed cases. Overall, there exists a strong positive correlation between PS and CG, with higher PS expected to yield better CG performance.

### 3.2 Featurizer: Parallelism Score Maximization

**Empirical Studies on Real-world Datasets.** To explore what characteristic of the models can achieve high PS representations, we evaluate multiple pre-trained models on the PACS, Office-Home and NICO datasets for PS and CG capabilities, also validating whether the above conclusions remain effective in more complex real-world scenarios. In specific, these datasets are broadly used in domain generalization, typically annotated with *class* and *domain* labels. We treat them as two target concepts and establish the CG tasks following Section 2.1. We select 65 pre-trained models with different architectures, sizes, training strategies, and training datasets from the `timm` library (Wightman, 2019). Through freezing their featurizers' parameters and conducting linear probing solely on representations, we evaluate the CG capabilities.

We separately record the $PS_{class}$ and $PS_{domain}$, $CG_{class}$ and $CG_{domain}$, yielding the following insights:

*(i). A positive correlation exists between PS and CG, and this conclusion still holds true even in real-world environments.* As depicted in Fig.5, Pearson's correlation ($\gamma$) and Spearman's rank correlation ($\rho$) coefficient both exceed 0.945, further emphasizing this robust correlation irrespective of dataset variations.

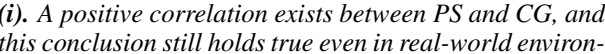

Figure 5: The $PS_{class}$ and $CG_{class}$ for different models across multiple datasets. Different colors represent different pre-trained models, different shapes represent different datasets.

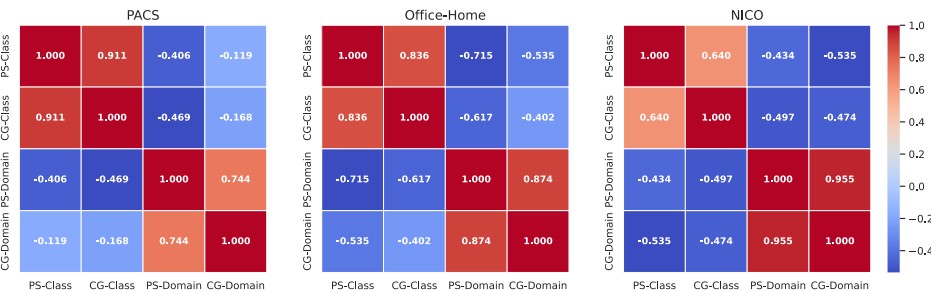

Figure 6: The Spearman correlation matrices of the PACS, Office-Home, and NICO datasets.

*(ii). Different models exhibit varying PS and CG capabilities across different concepts.* As indicated by the blue region in Fig.6, the $CG_{class}$ shows a negative correlation with the $PS_{domain}$ and $CG_{domain}$. This suggests that models with strong CG capabilities for *class* may have weaker CG capabilities for *domain*, indicating that the CG capabilities across different concepts are not necessarily correlated.

*(iii). The pre-training strategy of a model influences its PS and CG capability.* Models with randomly initialized weights inherently exhibit high PS for *domain*, showcasing the best $CG_{domain}$ performance but negligible $CG_{class}$ capability. Compared to supervised models, the self-supervised model DINO demonstrates a balanced performance on both *class* and *domain*, potentially because supervised models excessively focus on the *class* during the pre-training phase. For supervised models, increasing the amount of training data or scaling up the model size can enhance the $CG_{class}$ capability but does not improve the $CG_{domain}$ capability. Multimodal models like CLIP exhibit superior CG capabilities for both *class* and *domain*. Refer to the Appendix C for more details.

*(iv). Even the most advanced models struggle to exhibit strong CG capabilities across multiple concepts simultaneously.* So far, no model has demonstrated extremely strong CG capabilities in both *class* and *domain* aspects. Consequently, when it comes to more concepts, the model will face even greater difficulties. CG continues to be a highly complex and challenging task.

**Parallelism Score Maximization.** Existing models encounter difficulties in simultaneously possessing high PS across multiple concepts, thereby limiting their CG capabilities. Consequently, an intuitive approach is to utilize PS as a constraint and explicitly optimize it during the training process. Therefore, we propose the **P**arallelism **S**core **Max**imization (MaxPS) regularization, which encourages the representations to possess a highly parallel geometric structure. Since PS is a cosine function with an upper bound of 1, we achieve this by minimizing the difference between PS and 1.

Suppose that $T$ denotes the number of iterations. At each $t = 1, \ldots, T$ round, we get a batch of stochastic samples $\mathcal{D}_{tr}^t = \mathcal{X}_{tr}^t \times \mathcal{A}_{tr}^t \times \mathcal{B}_{tr}^t$. Following the mathematical definition in Eq.3, initially, we estimate the expected representation using batch samples as:

$$\hat{\boldsymbol{r}}^t(a, b) = \mathbb{E}_{\boldsymbol{x} \in \mathcal{X}_{tr}^t(a,b)}[g(\boldsymbol{x})] \tag{13}$$

When the batch size is large enough, this estimation approximates the actual expected value. If the batch size is limited, we can also employ an exponential smoothing method (Cutkosky & Orabona, 2019) to reduce the variance. According to Eq.8, the estimations of PS can be calculated as follows:

$$\hat{PS}(g_t; \mathcal{D}_{tr}^t) = \frac{1}{M_A^t + M_B^t} \sum_{a \neq a' \in \mathcal{A}_{tr}^t} \sum_{b \neq b' \in \mathcal{B}_{tr}^t} \left( \frac{1}{M_B^t} \cos \left\langle \hat{\boldsymbol{V}}^t(a \to a'|b), \hat{\boldsymbol{V}}^t(a \to a'|b') \right\rangle \right.$$
$$\left. + \frac{1}{M_A^t} \cos \left\langle \hat{\boldsymbol{V}}^t(b \to b'|a), \hat{\boldsymbol{V}}^t(b \to b'|a') \right\rangle \right) \tag{14}$$

where $\hat{\boldsymbol{V}}^t(a \to a'|b) = \hat{\boldsymbol{r}}^t(a, b) - \hat{\boldsymbol{r}}^t(a', b)$. Under the assumption of uniformly sampled concept transform vectors, this estimation can be proved unbiased as follows.

**Theorem 2.** *Assume that the concept transform vectors are uniformly sampled, and $\hat{\boldsymbol{r}}^t(a, b) \approx \overline{\boldsymbol{r}}(a, b)$, we have:* $\mathbb{E}[\hat{PS}(g_t; \mathcal{D}_{tr}^t)] \approx PS(g; \mathcal{D}_{tr})$.

Theorem 2 indicates that PS can be appropriately optimized. One might argue that uniformly sampled concept transform vectors may be too strong. In practice, we can reweight each pair of concept transform vectors to ensure equal influence, thereby potentially obtaining equivalent results.

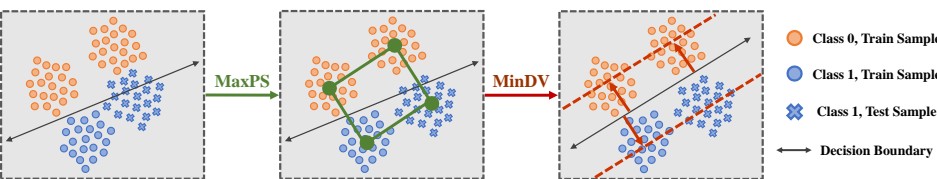

Figure 7: The Geometric Constraint framework.

By leveraging the PS estimation, we get the MaxPS regularized loss for the batch $t$ as:

$$\tilde{\mathcal{L}}_t = \mathcal{L}_{\mathcal{A}}(f_A^t, g_t; \mathcal{D}_{tr}^t) + \mathcal{L}_{\mathcal{B}}(f_B^t, g_t; \mathcal{D}_{tr}^t) + \beta(1 - \hat{PS}(g_t; \mathcal{D}_{tr}^t)) \tag{15}$$

and then perform gradient steps for descending $\tilde{\mathcal{L}}_t$ to update the model and obtain $f_A^{t+1}$, $f_B^{t+1}$ and $g_{t+1}$ until the parameter converges. The MaxPS regularization ensures that the representation models are enforced to optimize the accuracy and PS simultaneously, leading to better CG performance.

### 3.3 FRAMEWORK: GEOMETRIC CONSTRAINT

Based on the positive correlation between PS and CG capability, we introduce two regularizations: MinDV primarily constrains the geometric structure of the classifier, whereas MaxPS imposes constraints on the the representations extracted by the featurizer. By combining Eq.12 and Eq.15, we propose the **Geo**metric **Con**straint (GeoCon) method to enhance CG capability via PS and DV:

$$\min_{f_A, f_B, g} \mathbb{E}\Big[\mathcal{L}_{\mathcal{A}}(f_A, g) + \mathcal{L}_{\mathcal{B}}(f_B, g)\Big] + \alpha_A \hat{DV}(f_A, g) + \alpha_B \hat{DV}(f_B, g) + \beta(1 - \hat{PS}(g)) \tag{16}$$

where $\alpha_A$, $\alpha_B$ and $\beta$ denote the weights of the regularization terms. Fig.7 illustrates the workflow of GeoCon, where MaxPS encourages more parallelism among the centroids of the representations, while MinDV ensures a more equitable distribution of distances between sample points and decision boundary, thus conforming to this well-organized geometric structure. Refer to the Appendix A for the proof of the theorem and the Appendix B for detailed optimization steps.

## 4 EXPERIMENTS

### 4.1 SETUP

**Datasets.** As described in Section 2.1, the CG benchmark requires each sample to have at least two concept labels. Therefore, domain generalization datasets that concurrently annotate *class* and *domain* labels would serve as an feasible solution. We select the **PACS** (Li et al., 2017), **Office-Home** (Saenko et al., 2010), **DomainNet** (Peng et al., 2019), and **NICO** (He et al., 2021) datasets, considering their *class* and *domain* labels as two separate concepts, to construct our CG task, among which PACS, Office-Home, and DomainNet have concepts labeled as *content* and *style*, while NICO's concepts are labeled as *object* and *environment*. Refer to the Appendix D for more details.

Due to the significant disparity in the quantity of *classes* compared to *domains* in the Office-Home, DomainNet, and NICO datasets, there exists a substantial difficulty gap between classifying the two targets. To address this issue, we divide the datasets by *class* into several subsets, each containing a comparable number of *classes* as the number of *domains*, and record the average performance across these subsets. For each *class*, one *domain* is selected as the test set, while the remaining *domains* serve as the train set, ensuring that all *domains* have appeared in the train set.

For deeper analysis, we additionally leverage the synthetic dataset **3D Shapes** (Kim & Mnih, 2018), composed of 6 factors: *floor hue*, *wall hue*, *object hue*, *scale*, *shape*, and *orientation*. We identify *shape* and *object hue* as the concepts to be predicted, while regarding the other factors as noise, providing a reasonable simulation of the real world. We randomly sample 1000 images for each concept combination to construct our dataset. Half of the concept combinations are used for training, while the remaining combinations are employed for testing. Each image is flattened into a one-dimensional vector, and a three-layer fully connected neural network is employed, wherein the first two layers are shared by two concepts, serving as a featurizer. Subsequently, two linear classifiers are connected to classify *shape* and *object hue*, respectively.

Table 1: CG accuracy of models with different pre-training settings. Bold indicates the best result.

| Dataset | | PACS | | | Office-Home | | | DomainNet | | | NICO | | |
|---|---|---|---|---|---|---|---|---|---|---|---|---|---|
| Backbone | Method | class | domain | both | class | domain | both | class | domain | both | class | domain | both |
| ViT$_{1K}$ | LP | 0.4083 | 0.6834 | 0.2195 | 0.9155 | 0.2732 | 0.0789 | 0.3548 | 0.6052 | 0.1769 | 0.9313 | 0.2008 | 0.1521 |
| | FT | 0.7307 | 0.6655 | 0.4060 | 0.9127 | 0.3499 | 0.1553 | 0.7098 | 0.6125 | 0.3534 | 0.9323 | 0.2472 | 0.2366 |
| | GeoCon | 0.7896 | 0.6938 | **0.4804** | 0.9324 | 0.3962 | **0.2513** | 0.7103 | 0.6523 | **0.4352** | 0.9416 | 0.3228 | **0.3071** |
| ViT$_{21K}$ | LP | 0.5614 | 0.8433 | 0.4197 | 0.9465 | 0.2113 | 0.1690 | 0.5120 | 0.7513 | 0.3589 | 0.9550 | 0.3358 | 0.3047 |
| | FT | 0.7197 | 0.9085 | 0.6317 | 0.9437 | 0.2535 | 0.2000 | 0.6988 | 0.8230 | 0.5584 | 0.9629 | 0.3534 | 0.3233 |
| | GeoCon | 0.7884 | 0.9521 | **0.7409** | 0.9606 | 0.3299 | **0.2965** | 0.7765 | 0.8425 | **0.6327** | 0.9643 | 0.4420 | **0.4151** |
| DINO | LP | 0.4609 | 0.8245 | 0.2976 | 0.8366 | 0.2901 | 0.1549 | 0.4327 | 0.7450 | 0.2520 | 0.9323 | 0.3131 | 0.2630 |
| | FT | 0.5206 | 0.8245 | 0.3536 | 0.8394 | 0.3296 | 0.1775 | 0.4888 | 0.7552 | 0.3052 | 0.9341 | 0.3298 | 0.2782 |
| | GeoCon | 0.5493 | 0.8449 | **0.4043** | 0.8451 | 0.4231 | **0.2780** | 0.5075 | 0.8012 | **0.3721** | 0.8562 | 0.4212 | **0.3344** |
| CLIP | LP | 0.9384 | 0.9262 | 0.8645 | 0.9155 | 0.4620 | 0.3831 | 0.9012 | 0.8532 | 0.7865 | 0.9699 | 0.3984 | 0.3766 |
| | FT | 0.9234 | 0.9623 | 0.8857 | 0.9324 | 0.5803 | 0.5155 | 0.9026 | 0.8781 | 0.8014 | 0.9689 | 0.3789 | 0.3534 |
| | GeoCon | 0.9713 | 0.9910 | **0.9623** | 0.9114 | 0.6563 | **0.5775** | 0.9155 | 0.9352 | **0.8552** | 0.9768 | 0.4940 | **0.4819** |

Table 2: Comparison for CG accuracy of different methods on CLIP. Bold indicates the best result.

| Dataset | PACS | | | Office-Home | | | DomainNet | | | NICO | | | **Average** |
|---|---|---|---|---|---|---|---|---|---|---|---|---|---|
| Method | class | domain | both | class | domain | both | class | domain | both | class | domain | both | both |
| LP | 0.9384 | 0.9262 | 0.8645 | 0.9155 | 0.4620 | 0.3831 | 0.9012 | 0.8532 | 0.7865 | 0.9699 | 0.3984 | 0.3766 | 0.6027 |
| FT | 0.9234 | 0.9623 | 0.8857 | **0.9324** | 0.5803 | 0.5155 | 0.9026 | 0.8781 | 0.8014 | 0.9689 | 0.3789 | 0.3534 | 0.6390 |
| LP-FT | 0.9352 | 0.9713 | 0.9066 | 0.9218 | 0.6064 | 0.5216 | 0.9058 | 0.8810 | 0.8021 | 0.9702 | 0.3884 | 0.3615 | 0.6480 |
| WiSE-FT | 0.9399 | 0.9706 | 0.9109 | 0.9201 | 0.6255 | 0.5520 | 0.9094 | 0.9053 | 0.8233 | 0.9703 | 0.4402 | 0.4151 | 0.6753 |
| GeoCon w/o MaxPS | 0.9431 | 0.9761 | 0.9195 | 0.9147 | 0.6028 | 0.5433 | 0.9137 | 0.9184 | 0.8401 | 0.9712 | 0.4671 | 0.4439 | 0.6867 |
| GeoCon w/o MinDV | 0.9368 | 0.9623 | 0.8991 | 0.9195 | 0.6312 | 0.5524 | 0.9121 | 0.9136 | 0.8345 | 0.9717 | 0.4893 | 0.4763 | 0.6906 |
| GeoCon | **0.9713** | **0.9910** | **0.9623** | 0.9114 | **0.6563** | **0.5775** | **0.9155** | **0.9352** | **0.8552** | **0.9768** | **0.4940** | **0.4819** | **0.7192** |

**Baselines.** We implement the classifiers of the two concepts as single-layer linear classifiers. For the featurizer, we employ a ViT-Base-16 architecture and select models from the `timm` library (Wightman, 2019) under different training settings, including: **ViT$_{1K}$** (supervised training on ImageNet 1K), **ViT$_{21K}$** (supervised training on ImageNet 21K), **DINO** (self-supervised training) (Oquab et al., 2023), and **CLIP** (contrastive language-image pre-training) (Radford et al., 2021). CLIP tuning is an important research problem in transfer learning. To further validate the effectiveness of our approach, we select linear probing (**LP**), fine-tuning (**FT**), **LP-FT** (Kumar et al., 2022), and **WiSE-FT** (Wortsman et al., 2022) as baselines for comparison, where LP-FT does LP first and then FT after some epochs, WiSE-FT uses a weighted average of parameters before and after FT.

## 4.2 MAIN RESULTS

Tab.1 presents the *class* accuracy, *domain* accuracy, and both accuracy (classifying two concepts accurately simultaneously) on multiple datasets when utilizing pre-trained models under different training settings. Irrespective of the backbone employed, our method consistently demonstrates remarkably superior performance, thus substantiating the effectiveness of our approach. ViT$_{21K}$, compared to ViT$_{1K}$, demonstrates enhanced CG capability due to the increased training data volume. The supervised ViT$_{1K}$ and ViT$_{21K}$ result in higher performance in *class* accuracy, whereas the self-supervised DINO shows a more balanced performance in *class* accuracy and *domain* accuracy. CLIP continues to serve as a robust featurizer for CG tasks, surpassing other backbones significantly.

Tab.2 further analyzes the results of different algorithms when using CLIP as the backbone. Although WiSE-FT serves as a strong baseline for out-of-distribution generalization, our method still achieves state-of-the-art performance in this task. LP-FT shows a marginal improvement compared to LP and FT, suggesting that conventional methods focusing solely on train set accuracy may just enhance basic generalization but struggle to achieve strong CG capability. Furthermore, we conduct ablation studies to elucidate the importance of each component in our GeoCon framework. The results indicate that even independently utilizing one single component can lead to optimal performance. In comparison, MaxPS may be more critical since it primarily influences the featurizers with more parameters. Additionally, an observation reveals the presence of a "Buckets Effect" in CG tasks, wherein the both accuracy is more susceptible to concepts that are difficult to discriminate.

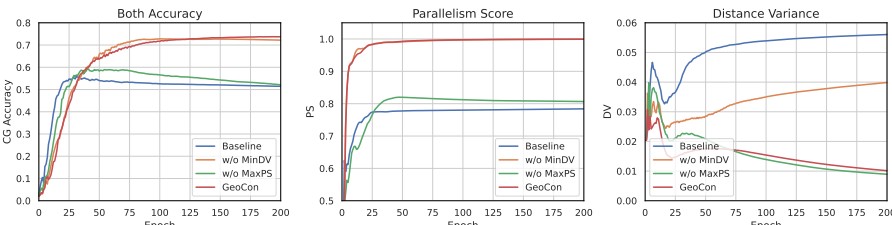

Figure 8: The training process on 3D Shapes Dataset.

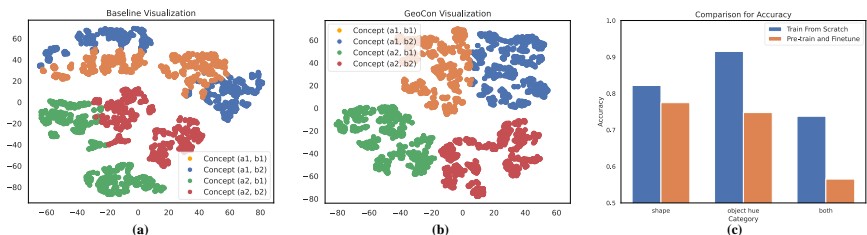

Figure 9: (a). t-SNE visualization when using baseline method. (b). t-SNE visualization when using GeoCon framework. (c). Comparison of the accuracy with (blue) and without (orange) pre-training.

### 4.3 DISCUSSIONS

We conduct a more comprehensive analysis on the 3D Shapes dataset, leveraging supervised learning without the addition of any regularization as the baseline method. Fig.8 illustrates the variation of metrics during the training process, wherein our approach demonstrates a significant improvement of 22.29% compared to the baseline. When MaxPS is not used, the model converges rapidly but experiences a decrease in accuracy after 25 epochs. It is noteworthy that the improvement in accuracy aligns precisely with the uplift of PS, while the decline in accuracy corresponds to the stagnation of PS. Maintaining consistency with MaxPS, the application of MinDV leads to lower DV and improved accuracy, without which DV will ascend.

Fig.9.(a) and Fig.9.(b) display the t-SNE visualization results. The representations of the baseline model still tend to cluster together but exhibit poor separability. In contrast, our GeoCon method produces highly separable representations characterized by distinct linear discriminability.

To investigate the impact of pre-training on CG, we initially train a featurizer supervised by *shape* solely and subsequently fine-tune it according to the CG setting. Surprisingly, the pre-trained models demonstrated lower accuracy compared to models trained from scratch, as depicted in Fig.9.(c). This could be attributed to biases introduced during the pre-training phase that are challenging to entirely eliminate during fine-tuning. This observation suggests that simply fine-tuning existing pre-trained models may not be efficacious for enhancing CG performance. The development of algorithms with stronger CG capability necessitates consideration from the pre-training stage.

## 5 CONCLUSIONS

**Conclusions.** We formally introduce the parallelism score from neuroscience to deep learning, revealing a strong positive correlation between it and compositional generalization from a geometry perspective. Our framework, GeoCon, consisting of MinDV for the classifier and MaxPS for the featurizer, aims to enhance CG capability by constraining the geometric structures of representations and forcing the decision boundaries to conform to the well-organized structure. Experiments show that GeoCon outperforms traditional approaches. This neuroscience-inspired representation mechanism may elucidate the fundamental nature of human-like intelligence in deep neural networks.

**Limitations.** Our GeoCon currently relies on aligned concepts, presenting challenges when scaling up to more complex and productive tasks. Furthermore, the computation of PS requires a substantial quantity of concept combinations inside each batch, which is of relative inefficiency. This issue could potentially be addressed by introducing a memory bank to mitigate computational complexity.

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

# A   PROOF FOR THEOREM

## A.1   PROOF FOR THEOREM 1

*Proof.* We shall begin with a 2-dimensional and 2-concept case. Assume that we have 4 points are $(a_1, a_2)$, $(b_1, b_2)$, $(c_1, c_2)$, $(d_1, d_2)$ with target $(y_1, y_2)$, $(y'_1, y_2)$, $(y_1, y'_2)$, $(y'_1, y'_2)$. If PS $= 1$, then we have:

$$\boldsymbol{V}_{y_1 \to y'_1, y_2} = (b_1 - a_1, b_2 - a_2) = (d_1 - c_1, d_2 - c_2) = \boldsymbol{V}_{y_1 \to y'_1, y_1}$$
$$\boldsymbol{V}_{y_1, y_2 \to y'_2} = (c_1 - a_1, c_2 - a_2) = (d_1 - b_1, d_2 - b_2) = \boldsymbol{V}_{y'_1, y_2 \to y'_2}$$

Since the representation space is linearly separable, we can easily know that the optimal classifier (considering the whole combination cases, hence have the best CG performance) for separating $y_1$ and $y'_1$ is:

$$y = \frac{c_2 - a_2}{c_1 - a_1}\left(x - \frac{a_1 + b_1}{2}\right) + \frac{a_2 + b_2}{2}$$

which is parallel to $\boldsymbol{V}_{y_1, y_2 \to y'_2}$ and $\boldsymbol{V}_{y'_1, y_2 \to y'_2}$. And the optimal classifier for separating $y_2$ and $y_2$ is:

$$y = \frac{b_2 - a_2}{b_1 - a_1}\left(x - \frac{a_1 + c_1}{2}\right) + \frac{a_2 + c_2}{2}$$

which is parallel to $\boldsymbol{V}_{y_1, y_2 \to y'_2}$ and $\boldsymbol{V}_{y'_1, y_2 \to y'_2}$.

Then, suppose the training points are $(a_1, a_2)$, $(b_1, b_2)$, $(c_1, c_2)$, the optimal solution of Eq.10 is exactly the same with the above optimal classifier that performs 100% accuracy when classifying $(d_1, d_2)$.

However, if four points are $A = (1, 1)$, $B = (2, 1)$, $C = (-1, 0)$, $D = (0, 0)$, then vanilla regression on $A$, $B$ and $C$ will have solution for separating $y_2$ and $y_2$:

$$x = \frac{3}{2}$$

which fail to classify $D$. We now prove that in the above case, Eq.10 yields a perfect model, while vanilla regression could fail.

In the more generalized case involving multiple concepts and higher dimensionality, the principle still comprises several instances of the 4-point cases. The generalization of our theoretical framework is readily achievable.                                                                            □

## A.2 Proof for Theorem 2

*Proof.* Since we know that $\hat{\boldsymbol{r}}^t(a,b) \approx \overline{\boldsymbol{r}}^t(a,b)$, we can get:

$$\hat{\boldsymbol{V}}^t_{a \to a',b} = \hat{\boldsymbol{r}}(a',b) - \hat{\boldsymbol{r}}(a,b) \approx \overline{\boldsymbol{r}}(a',b) - \overline{\boldsymbol{r}}(a,b) = \boldsymbol{V}^t_{a \to a',b},$$

$$\hat{\boldsymbol{V}}^t_{a,b \to b'} = \hat{\boldsymbol{r}}(a,b') - \hat{\boldsymbol{r}}(a,b) \approx \overline{\boldsymbol{r}}(a,b') - \overline{\boldsymbol{r}}(a,b) = \boldsymbol{V}^t_{a,b \to b'}.$$

For simplicity, we assume that $M_t = M^t_A = M^t_B$ in Eq.14 and $M = M_A = M_B$ in Eq.8. Due to the concept transform vectors are uniformly sampled, we have:

$$\mathbb{E}[\hat{\text{PS}}(g_t; \mathcal{D}^t_{tr})] = \mathbb{E}\left[\frac{1}{2M^2_t} \sum_{a \neq a' \in \mathcal{A}^t_{tr}} \sum_{b \neq b' \in \mathcal{B}^t_{tr}} \left( \cos\left\langle \hat{\boldsymbol{V}}^t_{a \to a',b}, \hat{\boldsymbol{V}}^t_{a \to a',b'} \right\rangle + \cos\left\langle \hat{\boldsymbol{V}}^t_{a,b \to b'}, \hat{\boldsymbol{V}}^t_{a',b \to b'} \right\rangle \right) \right]$$

$$\approx \mathbb{E}\left[\frac{1}{2M^2_t} \sum_{a \neq a' \in \mathcal{A}^t_{tr}} \sum_{b \neq b' \in \mathcal{B}^t_{tr}} \left( \cos\left\langle \boldsymbol{V}^t_{a \to a',b}, \boldsymbol{V}^t_{a \to a',b'} \right\rangle + \cos\left\langle \boldsymbol{V}^t_{a,b \to b'}, \boldsymbol{V}^t_{a',b \to b'} \right\rangle \right) \right]$$

$$= \frac{1}{2M^2} \sum_{a \neq a' \in \mathcal{A}_{tr}} \sum_{b \neq b' \in \mathcal{B}_{tr}} \left( \cos\left\langle \boldsymbol{V}^t_{a \to a',b}, \boldsymbol{V}^t_{a \to a',b'} \right\rangle + \cos\left\langle \boldsymbol{V}^t_{a,b \to b'}, \boldsymbol{V}^t_{a',b \to b'} \right\rangle \right)$$

$$= \text{PS}(g; \mathcal{D}_{tr}).$$

We thus end the proof. $\qquad\square$

# B    OPTIMIZATION STEPS FOR GEOCON

---

**Algorithm 1** Geometric Constraint for Compositional Generalization

---

1: **Input:** training dataset $\mathcal{D}_{tr} = \{(\boldsymbol{x}^{(i)}, \boldsymbol{a}^{(i)}, \boldsymbol{b}^{(i)})\}_{i=1}^N$, batch size $M$, learning rate $\eta$, training steps $T$, regularization weights $\alpha_A, \alpha_B, \beta$
2: **Output:** featurizer $g$, classifier $f_A$ for concept $A$, classifier $f_B$ for concept $B$ parameterized as $\theta$
3: **for** $t = 1$ to $T$ **do**
4:      get mini-batch data from $\mathcal{D}_{tr}$: $\mathcal{D}_{tr}^t = \{(\boldsymbol{x}^{(i)}, \boldsymbol{a}^{(i)}, \boldsymbol{b}^{(i)})\}_{i=1}^M$
5:      **for** $\forall (a,b) \in (\mathcal{A}_{tr}^t \times \mathcal{B}_{tr}^t)$ **do**
6:          calculate mean representations: $\hat{\boldsymbol{r}}^t(a,b)$ by Eq.3
7:      **end for**
8:      **for** $b = 1$ to $|\boldsymbol{b}|$ **do**
9:          **for** $\forall a \neq a' \in \mathcal{A}_{tr}^t$ **do**
10:              **if** $(a,b) \wedge (a',b) \in (\mathcal{A}_{tr}^t \times \mathcal{B}_{tr}^t)$ **then**
11:                  calculate concept $a \to a'$ transform vectors: $\hat{\boldsymbol{V}}^t(a \to a'|b)$ by Eq.4
12:              **end if**
13:          **end for**
14:      **end for**
15:      **for** $a = 1$ to $|\boldsymbol{a}|$ **do**
16:          **for** $\forall b \neq b' \in \mathcal{B}_{tr}^t$ **do**
17:              **if** $(a,b) \wedge (a,b') \in (\mathcal{A}_{tr}^t \times \mathcal{B}_{tr}^t)$ **then**
18:                  calculate concept $b \to b'$ transform vectors: $\hat{\boldsymbol{V}}^t(b \to b'|a)$ by Eq.5
19:              **end if**
20:          **end for**
21:      **end for**
22:      **for** $\forall a \neq a' \in \mathcal{A}_{tr}^t$ **do**
23:          **for** $\forall b \neq b' \in \mathcal{B}_{tr}^t$ **do**
24:              **if** $(a,b) \wedge (a,b') \wedge (a',b) \wedge (a',b') \in (\mathcal{A}_{tr}^t \times \mathcal{B}_{tr}^t)$ **then**
25:                  calculate parallelism score: $\hat{\text{PS}}(g_t; \mathcal{D}_{tr}^t)$ by Eq.8
26:              **end if**
27:          **end for**
28:      **end for**
29:      **for** $i = 1$ to $M$ **do**
30:          calculate distances: $\dfrac{\|f_A \circ g(\boldsymbol{x}^{(i)})\|}{\|\boldsymbol{w}_A\|_2}$ and $\dfrac{\|f_B \circ g(\boldsymbol{x}^{(i)})\|}{\|\boldsymbol{w}_B\|_2}$
31:          calculate cross-entropy loss: $\mathcal{L}_A(f_A^t, g_t; \mathcal{D}_{tr}^t)$ and $\mathcal{L}_B(f_B^t, g_t; \mathcal{D}_{tr}^t)$
32:      **end for**
33:      calculate distance variance: $\hat{\text{DV}}_A(f_A^t, g_t; \mathcal{D}_{tr}^t)$ and $\hat{\text{DV}}_B(f_B^t, g_t; \mathcal{D}_{tr}^t)$ by Eq.10
34:      calculate total loss: $\tilde{\mathcal{L}}_t = \mathcal{L}_A + \mathcal{L}_B + \alpha_A \hat{\text{DV}}_A + \alpha_B \hat{\text{DV}}_B + \beta(1 - \hat{\text{PS}})$ by Eq.16
35:      update parameters by stochastic gradient descent: $\theta_{t+1} \leftarrow \theta_t - \eta \dfrac{\partial \tilde{\mathcal{L}}_t}{\partial \theta_t}$
36: **end for**

# C  THE PS AND CG ACCURACY OF DIFFERENT PRE-TRAINED MODELS

Table 3: The PS and CG Accuracy of different pre-trained models on the PACS dataset.

| Model | PS-class | CG-class | PS-domain | CG-domain |
|---|---|---|---|---|
| resnet18_random | -0.0017 | 0 | 0.9929 | 1 |
| resnet18_tv_in1k | 0.3697 | 0.3805 | 0.5864 | 0.8499 |
| resnet18_fb_ssl_yfcc100m_ft_in1k | 0.3951 | 0.3542 | 0.6127 | 0.8591 |
| resnet18_fb_swsl_ig1b_ft_in1k | 0.5225 | 0.5012 | 0.5724 | 0.8569 |
| resnet34_tv_in1k | 0.3581 | 0.4055 | 0.574 | 0.8533 |
| resnet50_random | -0.0024 | 0 | 0.9946 | 1 |
| resnet50_tv2_in1k | 0.3354 | 0.4073 | 0.5663 | 0.8526 |
| resnet50_fb_ssl_yfcc100m_ft_in1k | 0.4263 | 0.3418 | 0.6004 | 0.8682 |
| resnet50_fb_swsl_ig1b_ft_in1k | 0.6123 | 0.6333 | 0.5608 | 0.8711 |
| resnet101_tv2_in1k | 0.3256 | 0.3844 | 0.5751 | 0.8206 |
| resnet152_tv2_in1k | 0.3325 | 0.383 | 0.563 | 0.812 |
| wide_resnet50_2_tv2_in1k | 0.3221 | 0.3 | 0.5606 | 0.8504 |
| wide_resnet101_2_tv2_in1k | 0.3171 | 0.4144 | 0.586 | 0.8411 |
| vit_small_patch8_224_dino | 0.4028 | 0.5302 | 0.6058 | 0.8552 |
| vit_small_patch16_224_augreg_in1k | 0.3432 | 0.3062 | 0.5409 | 0.8248 |
| vit_small_patch16_224_augreg_in21k | 0.4456 | 0.3945 | 0.6144 | 0.9193 |
| vit_small_patch16_224_augreg_in21k_ft_in1k | 0.4629 | 0.4464 | 0.5911 | 0.9064 |
| vit_small_patch16_224_dino | 0.3678 | 0.3412 | 0.627 | 0.8852 |
| vit_small_patch32_224_augreg_in21k | 0.3778 | 0.3477 | 0.5804 | 0.8894 |
| vit_small_patch32_224_augreg_in21k_ft_in1k | 0.3747 | 0.3717 | 0.5694 | 0.8434 |
| vit_base_patch16_224_random | -0.0021 | 0 | 0.9493 | 1 |
| vit_base_patch16_224_augreg_in1k | 0.3813 | 0.3497 | 0.5481 | 0.8089 |
| vit_base_patch16_224_augreg_in21k | 0.4876 | 0.5275 | 0.6244 | 0.9364 |
| vit_base_patch16_224_augreg_in21k_ft_in1k | 0.5514 | 0.5543 | 0.572 | 0.8588 |
| vit_base_patch16_224_dino | 0.3854 | 0.3928 | 0.6108 | 0.8767 |
| vit_base_patch16_224_mae | 0.1319 | 0.132 | 0.5857 | 0.8246 |
| vit_base_patch16_clip_224_laion2b | 0.7491 | 0.8638 | 0.7334 | 0.9719 |
| vit_base_patch16_clip_224_laion2b_ft_in1k | 0.6145 | 0.6976 | 0.5328 | 0.8891 |
| vit_base_patch16_clip_224_laion2b_ft_in12k_in1k | 0.535 | 0.5601 | 0.4925 | 0.8523 |
| vit_base_patch16_clip_224_openai | 0.7369 | 0.8035 | 0.6761 | 0.9287 |
| vit_base_patch16_clip_224_openai_ft_in1k | 0.6225 | 0.7693 | 0.4733 | 0.8417 |
| vit_base_patch16_clip_224_openai_ft_in12k_in1k | 0.5292 | 0.6341 | 0.4515 | 0.833 |
| vit_base_patch32_224_random | -0.0023 | 0 | 0.9443 | 1 |
| vit_base_patch32_224_augreg_in1k | 0.3447 | 0.281 | 0.5534 | 0.8168 |
| vit_base_patch32_224_augreg_in21k | 0.4187 | 0.4261 | 0.6191 | 0.8761 |
| vit_base_patch32_224_augreg_in21k_ft_in1k | 0.4432 | 0.4463 | 0.5786 | 0.8837 |
| vit_base_patch32_clip_224_laion2b | 0.7293 | 0.769 | 0.7193 | 0.9794 |
| vit_base_patch32_clip_224_laion2b_ft_in12k_in1k | 0.5185 | 0.5523 | 0.4712 | 0.8698 |
| vit_base_patch32_clip_224_laion2b_ft_in1k | 0.5537 | 0.5642 | 0.5367 | 0.8507 |
| vit_large_patch16_224_random | -0.0017 | 0 | 0.9591 | 1 |
| vit_large_patch16_224_augreg_in21k | 0.519 | 0.514 | 0.6279 | 0.9267 |
| vit_large_patch16_224_augreg_in21k_ft_in1k | 0.6149 | 0.7023 | 0.5985 | 0.901 |
| vit_large_patch16_224_mae | 0.3074 | 0.1342 | 0.7857 | 0.9129 |

Table 4: The PS and CG Accuracy of different pre-trained models on the Office-Home dataset.

| Model | PS-class | CG-class | PS-domain | CG-domain |
|---|---|---|---|---|
| resnet18_random | 0.0161 | 0 | 0.9836 | 1 |
| resnet18_tv_in1k | 0.6987 | 0.5559 | 0.2555 | 0.5614 |
| resnet18_fb_ssl_yfcc100m_ft_in1k | 0.6864 | 0.5463 | 0.2697 | 0.5828 |
| resnet18_fb_swsl_ig1b_ft_in1k | 0.7135 | 0.5805 | 0.2755 | 0.562 |
| resnet34_tv_in1k | 0.7136 | 0.5832 | 0.2605 | 0.5532 |
| resnet50_random | 0.0052 | 0 | 0.9864 | 1 |
| resnet50_tv2_in1k | 0.6928 | 0.634 | 0.2068 | 0.5092 |
| resnet50_fb_ssl_yfcc100m_ft_in1k | 0.6997 | 0.6165 | 0.265 | 0.5774 |
| resnet50_fb_swsl_ig1b_ft_in1k | 0.7273 | 0.6706 | 0.2637 | 0.5846 |
| resnet101_tv2_in1k | 0.71 | 0.641 | 0.2266 | 0.4892 |
| resnet152_tv2_in1k | 0.7141 | 0.6467 | 0.2068 | 0.5004 |
| wide_resnet50_2_tv2_in1k | 0.6883 | 0.625 | 0.1963 | 0.5056 |
| wide_resnet101_2_tv2_in1k | 0.7085 | 0.6609 | 0.2115 | 0.4888 |
| vit_small_patch8_224_dino | 0.6678 | 0.594 | 0.3636 | 0.6542 |
| vit_small_patch16_224_augreg_in1k | 0.7231 | 0.6369 | 0.1632 | 0.5289 |
| vit_small_patch16_224_augreg_in21k | 0.7143 | 0.7102 | 0.2644 | 0.6167 |
| vit_small_patch16_224_augreg_in21k_ft_in1k | 0.7568 | 0.7141 | 0.2421 | 0.6003 |
| vit_small_patch16_224_dino | 0.6454 | 0.5095 | 0.3521 | 0.6715 |
| vit_small_patch32_224_augreg_in21k | 0.7115 | 0.6799 | 0.2402 | 0.5897 |
| vit_small_patch32_224_augreg_in21k_ft_in1k | 0.7378 | 0.6772 | 0.2182 | 0.5654 |
| vit_base_patch16_224_random | 0.0051 | 0 | 0.86 | 1 |
| vit_base_patch16_224_augreg_in1k | 0.7437 | 0.6238 | 0.1542 | 0.4942 |
| vit_base_patch16_224_augreg_in21k | 0.7019 | 0.7528 | 0.269 | 0.6549 |
| vit_base_patch16_224_augreg_in21k_ft_in1k | 0.7881 | 0.7672 | 0.2145 | 0.6212 |
| vit_base_patch16_224_dino | 0.6549 | 0.57 | 0.3472 | 0.6412 |
| vit_base_patch16_224_mae | 0.4166 | 0.097 | 0.32 | 0.5989 |
| vit_base_patch16_clip_224_laion2b | 0.7106 | 0.7438 | 0.4751 | 0.8177 |
| vit_base_patch16_clip_224_laion2b_ft_in1k | 0.8073 | 0.7896 | 0.1993 | 0.5451 |
| vit_base_patch16_clip_224_laion2b_ft_in12k_in1k | 0.7951 | 0.7867 | 0.1775 | 0.54 |
| vit_base_patch16_clip_224_openai | 0.7039 | 0.6471 | 0.4516 | 0.7672 |
| vit_base_patch16_clip_224_openai_ft_in1k | 0.7937 | 0.7489 | 0.1831 | 0.5408 |
| vit_base_patch16_clip_224_openai_ft_in12k_in1k | 0.784 | 0.7741 | 0.1441 | 0.4959 |
| vit_base_patch32_224_random | 0.0026 | 0 | 0.8676 | 1 |
| vit_base_patch32_224_augreg_in1k | 0.7135 | 0.6194 | 0.1566 | 0.5114 |
| vit_base_patch32_224_augreg_in21k | 0.677 | 0.6995 | 0.2499 | 0.6128 |
| vit_base_patch32_224_augreg_in21k_ft_in1k | 0.729 | 0.7089 | 0.2156 | 0.5676 |
| vit_base_patch32_clip_224_laion2b | 0.6972 | 0.7091 | 0.4539 | 0.7953 |
| vit_base_patch32_clip_224_laion2b_ft_in12k_in1k | 0.7721 | 0.7539 | 0.1671 | 0.5093 |
| vit_base_patch32_clip_224_laion2b_ft_in1k | 0.7741 | 0.7277 | 0.2166 | 0.5577 |
| vit_large_patch16_224_random | 0.0014 | 0 | 0.884 | 1 |
| vit_large_patch16_224_augreg_in21k | 0.7122 | 0.76 | 0.2768 | 0.6472 |
| vit_large_patch16_224_augreg_in21k_ft_in1k | 0.8136 | 0.7998 | 0.2303 | 0.61 |
| vit_large_patch16_224_mae | 0.5065 | 0.1118 | 0.4222 | 0.6601 |

Table 5: The PS and CG Accuracy of different pre-trained models on the NICO dataset.

| Model | PS-class | CG-class | PS-domain | CG-domain |
|---|---|---|---|---|
| resnet18_random | -0.0054 | 0 | 0.9897 | 1 |
| resnet18_tv_in1k | 0.8006 | 0.6717 | 0.4011 | 0.4513 |
| resnet18_fb_ssl_yfcc100m_ft_in1k | 0.8051 | 0.7289 | 0.4053 | 0.4573 |
| resnet18_fb_swsl_ig1b_ft_in1k | 0.8109 | 0.7523 | 0.3912 | 0.4388 |
| resnet34_tv_in1k | 0.8113 | 0.7188 | 0.3819 | 0.4372 |
| resnet50_random | -0.0045 | 0 | 0.9913 | 1 |
| resnet50_tv2_in1k | 0.8014 | 0.7859 | 0.3164 | 0.3652 |
| resnet50_fb_ssl_yfcc100m_ft_in1k | 0.8168 | 0.8115 | 0.3707 | 0.4154 |
| resnet50_fb_swsl_ig1b_ft_in1k | 0.826 | 0.8213 | 0.3603 | 0.4153 |
| resnet101_tv2_in1k | 0.7966 | 0.7756 | 0.3287 | 0.3609 |
| resnet152_tv2_in1k | 0.7946 | 0.7531 | 0.3271 | 0.3749 |
| wide_resnet50_2_tv2_in1k | 0.7996 | 0.7827 | 0.3005 | 0.3642 |
| wide_resnet101_2_tv2_in1k | 0.7961 | 0.7816 | 0.2914 | 0.3678 |
| vit_small_patch8_224_dino | 0.8159 | 0.8305 | 0.3382 | 0.4306 |
| vit_small_patch16_224_augreg_in1k | 0.8206 | 0.7719 | 0.2464 | 0.351 |
| vit_small_patch16_224_augreg_in21k | 0.8135 | 0.8385 | 0.4044 | 0.4613 |
| vit_small_patch16_224_augreg_in21k_ft_in1k | 0.8409 | 0.8418 | 0.3805 | 0.4273 |
| vit_small_patch16_224_dino | 0.782 | 0.7706 | 0.3601 | 0.4564 |
| vit_small_patch32_224_augreg_in21k | 0.79 | 0.7651 | 0.3861 | 0.4743 |
| vit_small_patch32_224_augreg_in21k_ft_in1k | 0.8109 | 0.7705 | 0.3657 | 0.4583 |
| vit_base_patch16_224_random | -0.0018 | 0 | 0.9084 | 1 |
| vit_base_patch16_224_augreg_in1k | 0.8262 | 0.7621 | 0.238 | 0.3502 |
| vit_base_patch16_224_augreg_in21k | 0.7935 | 0.8639 | 0.3827 | 0.4567 |
| vit_base_patch16_224_augreg_in21k_ft_in1k | 0.8575 | 0.8711 | 0.363 | 0.4124 |
| vit_base_patch16_224_dino | 0.7989 | 0.7964 | 0.3495 | 0.436 |
| vit_base_patch16_224_mae | 0.6242 | 0.2528 | 0.7944 | 0.5234 |
| vit_base_patch16_clip_224_laion2b | 0.8265 | 0.8786 | 0.4397 | 0.4998 |
| vit_base_patch16_clip_224_laion2b_ft_in1k | 0.8258 | 0.8644 | 0.2296 | 0.3262 |
| vit_base_patch16_clip_224_laion2b_ft_in12k_in1k | 0.8269 | 0.8683 | 0.2171 | 0.3369 |
| vit_base_patch16_clip_224_openai | 0.8329 | 0.8653 | 0.4059 | 0.4673 |
| vit_base_patch16_clip_224_openai_ft_in1k | 0.8303 | 0.8641 | 0.1984 | 0.333 |
| vit_base_patch16_clip_224_openai_ft_in12k_in1k | 0.8219 | 0.877 | 0.2143 | 0.3453 |
| vit_base_patch32_224_random | 0.0038 | 0 | 0.9015 | 1 |
| vit_base_patch32_224_augreg_in1k | 0.7887 | 0.699 | 0.2814 | 0.4104 |
| vit_base_patch32_224_augreg_in21k | 0.7631 | 0.8044 | 0.384 | 0.4683 |
| vit_base_patch32_224_augreg_in21k_ft_in1k | 0.8088 | 0.8187 | 0.3572 | 0.4399 |
| vit_base_patch32_clip_224_laion2b | 0.8147 | 0.8384 | 0.4445 | 0.5143 |
| vit_base_patch32_clip_224_laion2b_ft_in12k_in1k | 0.8087 | 0.8475 | 0.2305 | 0.3434 |
| vit_base_patch32_clip_224_laion2b_ft_in1k | 0.7972 | 0.8231 | 0.2537 | 0.3618 |
| vit_large_patch16_224_random | -0.0014 | 0 | 0.9192 | 1 |
| vit_large_patch16_224_augreg_in21k | 0.7883 | 0.8746 | 0.3744 | 0.476 |
| vit_large_patch16_224_augreg_in21k_ft_in1k | 0.8641 | 0.8838 | 0.3553 | 0.4164 |
| vit_large_patch16_224_mae | 0.6728 | 0.3345 | 0.5714 | 0.5421 |

# D  DATASET INFORMATION

Every domain generalization dataset is divided into several groups by *domain*, and each domain group is divided into several categories by *class*, while categories across different *domains* are the same. It means that every image in the dataset will have two labels, a *class* label and a *domain* label. Here, we have presented examples of the PACS dataset in Fig.10.

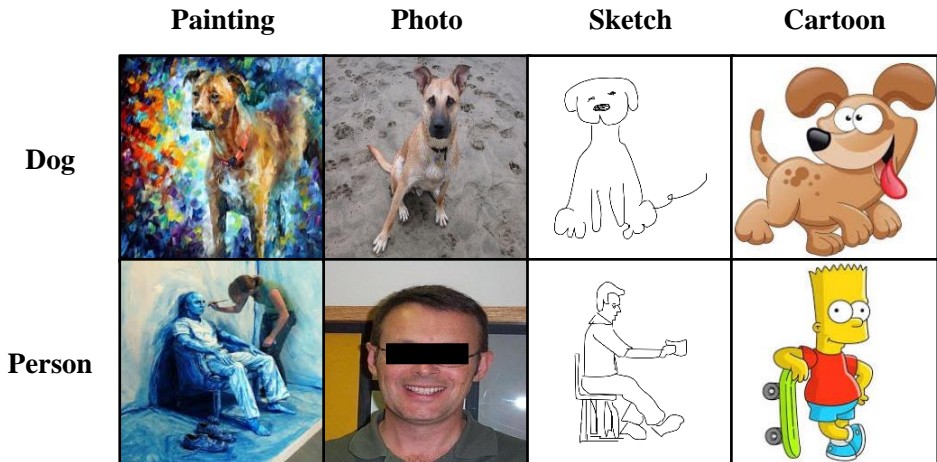

Figure 10: Examples of the PACS domain generalization dataset. Here, *dog* and *person* are *classes*, while *painting*, *photo*, *sketch* and *cartoon* are *domains*.

Shapes3D is a dataset of 3D shapes procedurally generated from 6 ground truth independent latent factors. These factors are *floor hue* (10 values), *wall hue* (10 values), *object hue* (10 values), *scale* (8 values), *shape* (4 values), and *orientation* (15 values). All possible combinations of these latents are present exactly once, generating $N = 480,000$ total images. We identify *shape* and *object hue* as the concepts to be predicted, while regarding the other factors as noise, as demonstrated in Fig.11.

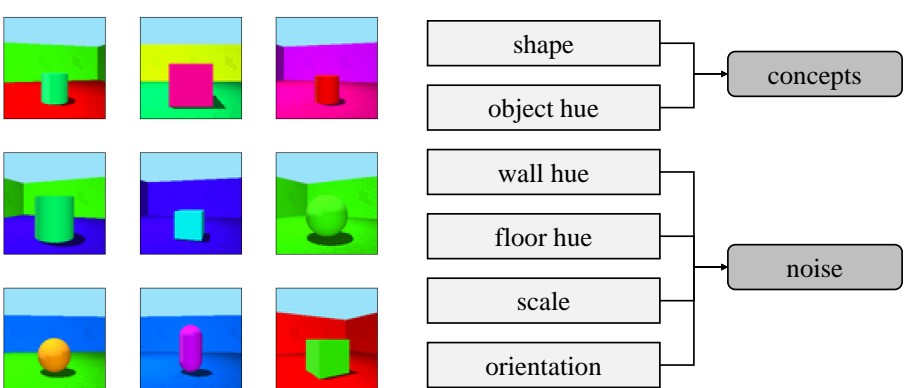

Figure 11: Examples of Shapes3D dataset.

# E  MORE ABLATION STUDY RESULTS

## E.1  CONCEPT FACTOR QUANTITY

To facilitate comprehension and maintain simplicity, we just discuss the situation when the concept factor is two. Here we will add detailed information when there are more than two concepts: When the quantity of concept factors is N, there are N targets to be predicted. For the featurizer, they will share a public one as a visual encoder to extract features. For classifiers, we arrange N independent linear functions to predict N concepts. For MinDV regularization, we calculate distance variance within N concepts, indicating that N regularization terms will exist for N concepts. For MaxPS regularization, to compute PS, two groups of concept factors are needed, and we regard other concept factors as noise, like what we do in the Shapes3D dataset. We traverse all possible compositions of any two concepts and sum them up to get the final result.

We test our method when there are three concepts in the Shapes3D dataset: *object hue*, *shape*, and *wall hue*. The separation of train set and test set brings into correspondence with Section 4.1. The results of different methods are shown in Tab.6, validating our GeoCon method's effectiveness.

Table 6: CG accuracy when the quantity of concept factors is three in Shapes3D dataset. Bold indicates the best result.

| Method | Acc-object hue | Acc-wall hue | Acc-shape | Acc-all |
|---|---|---|---|---|
| baseline | 0.6910 | 0.7531 | 0.5434 | 0.2853 |
| w/o MaxPS | 0.7018 | 0.7271 | 0.5849 | 0.3145 |
| w/o MinDV | 0.8187 | 0.8566 | 0.7023 | 0.4839 |
| GeoCon | **0.8405** | **0.8842** | **0.7341** | **0.5241** |

## E.2  BATCH SIZE

To get an absolutely accurate PS, it needs to calculate all samples of the dataset. According to Eq.14, the minimal sample quantity required to calculate PS is 4 with the target concepts of $(a_1, b_1)$, $(a_1, b_2)$, $(a_2, b_1)$, and $(a_2, b_2)$.

As stated in Section 3.2, we hope to generate an estimation of PS as accurately as possible. The batch size depends on the specific dataset, including the number of concept factors and the number of target categories. More concept factors and target categories imply a larger batch size. In practice, we set the batch size as 256. Compared to the domain quantity of no more than 6 and the class quantity of no more than 7, the batch size is big enough to generate an accurate estimation. However, when the batch size is limited, we can employ an exponential smoothing method to reduce the estimation variance, thus resolving the challenge.

To demonstrate that our method remains effective even with smaller batch size, we conduct experiments with batch size 32 as demonstrated in Tab.7. Compared to GeoCon with batch size 256, there was only a negligible drop in performance, while outperforming the FT baseline much.

Table 7: CG accuracy in the real-world datasets under different settings of batch size.

| Method | PACS | Office-Home | DomainNet | NICO |
|---|---|---|---|---|
| FT | 0.8857 | 0.5155 | 0.8014 | 0.3534 |
| GeoCon-32 | 0.9557 | 0.5521 | 0.8414 | 0.4695 |
| GeoCon-256 | 0.9623 | 0.5775 | 0.8552 | 0.4819 |

### E.3 OTHER BASELINE

We try to understand the essence of Ito et al. (2022) and validate it in our tasks. Specifically, their method doesn't random sample from mixing all the composition data. Instead, they try to learn a single concept first and then extend to some new compositions. The core of their algorithm is to start from simple settings and then gradually expand to complex compositions. Since their method is not open-source, we refer to Appendix 7 and 8 in Ito et al. (2022) and implement their algorithm through the following approach: First, we conduct training on two randomly selected groups of concepts (on the four corresponding compositions), and then gradually expand to new compositions one by one until all the compositions in the train set have been trained. The results are demonstrated in Tab.8, where primitives pre-training (PPT) refers to their method. We find that, compared with FT, PPT shows an improvement when FT doesn't work well, but is still inferior to our GeoCon method.

Table 8: The comparison between GeoCon and PPT across multiple datasets. We present the accuracy of predicting two concepts correctly at the same time. Bold indicates the best result.

| Method | PACS | Office-Home | DomainNet | NICO | Shapes3D |
|---|---|---|---|---|---|
| FT / from Scratch | 0.8857 | 0.5155 | 0.8014 | 0.3534 | 0.5145 |
| PPT | 0.8838 | 0.5324 | 0.8030 | 0.4012 | 0.6094 |
| GeoCon | **0.9623** | **0.5775** | **0.8552** | **0.4819** | **0.7374** |

