# OpenReview forum: "GeoCon: Compositional Generalization Through Geometric Constraints on Representation Structure"
_ICLR.cc/2025/Conference — ICLR 2025 Conference Withdrawn Submission_

### Official Review · Reviewer_9Z46 · 2024-10-19

**Soundness:** 4
**Presentation:** 3
**Contribution:** 3
**Rating:** 8
**Confidence:** 4

**Summary:**

Discusses parallelism score (PS), from neuroscience, as a proxy measure of compositional generalization (CG) in artificial NNs.

Shows that their measure of PS correlates with CG across multiple datasets and models.

Shows that minimizing an approximation to PS, as well as an additional loss term that aims to keep points a constant distance from the classifier's decision boundary, improves CG.

**Strengths:**

Compositional generalization is an interesting problem.

Parallelism score seems to make sense theoretically and is generally well-explained. It is nice that it can be optimized directly, as opposed to most measures of disentanglement.

Very convincing experimental results, both in the correlation of PS and CG, and in the improvement in CG from the additional loss terms.

**Weaknesses:**

When introducing PS score, it's not clear to me how exactly it relates to the neuroscience measure of Bernardi et al. (2020). Presumably the latter was not able to calculate cosine similarity of different activation vectors.

Could you give more details about the datasets? what types of objects are they classifying? what are some examples of values for ‘domain’ and ‘class’?

If I understand correctly, the experiments in Section 3.2 form a test set of points that have a novel combination of class and domain, and then ‘CG class’ is the fraction of these points for which the class is predicted correctly, and the same for ‘CG domain’. In that case, I wonder how getting both correct correlates with PS score, especially in light of the fact that CG class and CG domain are negatively correlated. This is included in Tables 1 and 2, but I’d recommend saying something about it in discussing Figure 6 too. Short of putting this in as another row, ‘CG both’, in the plots of Figure 6, it would still be helpful just to mention this alternative in the text.

Section 3.2, point (iv) is confusing to me. There are only two concepts here, class and domain, what would it mean to have CG across multiple concepts?

line 364: “we can also employ an exponential smoothing method”--does that mean you do employ it?

The approximation to PS seems to require a certain amount of diversity in the batch, what batch size do you use?

The way you measure CG is very similar to Xu et al. (2022) and Mahon et al. (2024).

Line 050: “there is currently no evidence that explicitly decoupling input compositional factors substantially improves the learning efficiency or generalization capacity”--this is too strong a statement in my view. On whether there is a connection between disentanglement and CG, there are papers on both sides: those you cite claim there is not, while Higgins et al. (2016) and Esmaeli et al (2019) both claim that their methods for disentanglement show evidence of facilitating CG, and Mahon et al. (2024) claims to find a correlation between disentanglement and CG across various models and datasets. If you are to take a side on that issue, you could measure disentanglement on the embeddings you take from the frozen models in Section 3.2, and see if it correlates with CG and maybe also PS. That would be an interesting addition to your work.

**Refs**

Higgins et al. (2016) "beta-vae: Learning basic visual concepts with a constrained variational framework."

Esmaeli et al. (2019) "Structured Disentangled Representations"

Mahon et al. (2024) "Correcting Flaws in Common Disentanglement Metrics"


**Minor points**

103: “Featurizer" --> "The featurizer”

134: "the linear function" --> “a linear function”

if g and D are on the LHS of Eq (8), shouldn’t they also be on the LHS of (7) and (6)? i.e. $PS(a)$, should be $PS(a;g,D)$, or you could just remove g and D from (8) taking them to be implicit everywhere

What are $L_a$ and $L_B$ in Eq (10)? (stated earlier that they’re CE, but helpful to repeat especially before the them.)

Does 'accuracy' in Thm. 1 refer to train set accuracy?

**Questions:**

Could you please reply to the points I mentioned in 'weaknesses' above? (most are more questions that weaknesses) Thanks.

---

> ### Author Response · Authors · 2024-11-20
>
> Dear Reviewer 9Z46,
>
> Thank you for your positive and thoughtful feedback and look into every detail of our work. We appreciate that you found our proposed method well-explained and experiments convincing. We summarize the mentioned concerns and hope the corresponding comments address your concerns, we would be grateful if you could consider increasing the review score.
>
> >Q1. When introducing PS score, it's not clear to me how exactly it relates to the neuroscience measure of Bernardi et al. (2020).
>
> A1. According to Bernardi et al. (Cell 2020), they develop a measure based on angles of coding directions to characterize the geometry of neural representations of variables in the firing rate space. First, the neural activity of individual neurons in the hippocampus, dorsolateral prefrontal cortex, and anterior cingulate cortex is recorded while monkeys perform a serial-reversal learning task. Neurons that have at least ten trials for each experimental condition meeting the selection criteria are retained. Then, the spike count distribution of each neuron is z-scored separately, which helps in standardizing the neural activity data before further analysis. Finally, within a chosen time bin, the activity of each neuron across all conditions is averaged across trials to obtain the firing rate patterns for each condition. These firing rate patterns are used to calculate the PS. You can refer to Pages 19 and 23 for more information.
>
> In our setting, we are motivated by their main idea: "A high PS indicates that the coding directions for a variable are more parallel, which is related to the ability of a neural representation to support generalization." We regard different concept combinations as different conditions and calculate the PS, to verify whether it holds true in DNNs.
>
> >Q2. Could you give more details about the datasets? What types of objects are they classifying? What are some examples of values for ‘domain’ and ‘class’?
>
> A2. In real-world datasets, each dataset is divided into several groups by domain, and each domain group is divided into several categories by class, while categories across different domains are the same. It means that every image in the dataset will have two labels, a class label and a domain label. For example, as illustrated in Figure 1 (b), in the PACS dataset, classes include Horse, Giraffe, Elephant, Dog, and Person, and domains include Painting, Sketch, Photo, and Cartoon. You can find some "sketch elephant" images in the dataset.
>
> For easier comprehension, we will give some instances of datasets in the Appendix D in the updated version.
>
> >Q3. In that case, I wonder how getting both correct correlates with PS score, especially in light of the fact that CG class and CG domain are negatively correlated.
>
> A3. Thanks for your sincere advice. It should be clarified that we are only concerned with the capability to classify a single concept under the CG setting. This is because the summation operation in Equation 8 will result in the loss of abundant information. For example, when a quadrilateral is a trapezoid, the PS of two pairs of its sides may be 1, while the PS of the other two pairs of sides may be 0 or even negative. The summation operation will overlook such examples that are rich in information.
>
> Nevertheless, we are also very grateful for your suggestions, which can help make our paper clearer and more understandable. Since this description also needs to present the results of CG-both and PS-average, which requires quite a lot of updates in terms of layout, we are considering supplementing this part of the content in the camera-ready version.
>
> >Q4. There are only two concepts here, class and domain, what would it mean to have CG across multiple concepts?
>
> A4. Thank you for your careful reading and sorry for the misleading. Here we would like to convey that, in the case of dealing with two concepts, the existing models already find it very difficult to perform well simultaneously. Consequently, when it comes to more concepts, it will be even more challenging, and the existing models will also struggle to exhibit strong CG capabilities.
>
> This was our imprecise expression and we have already corrected it in the updated version.

---

> ### Author Response · Authors · 2024-11-20
>
> >Q5. Line 364: “we can also employ an exponential smoothing method”--does that mean you do employ it? The approximation to PS seems to require a certain amount of diversity in the batch, what batch size do you use?
>
> A5. In practice, we set the batch size as 256. Compared to the domain quantity of no more than 6 and the class quantity of no more than 7, the batch size is big enough to generate an accurate estimation. So we do not use the exponential smoothing method. Here we just mention it to highlight that through this method, our proposed GeoCon also works well, though the batch size is limited.
>
> To demonstrate that our method remains effective even with smaller batch size, we conduct experiments with batch size 32. Compared to GeoCon with batch size 256, there was only a negligible drop in performance, while outperforming the FT baseline much.
>
> |   Method   |  PACS  |  Office-Home | DomainNet |  NICO  |
> |:----------:|:------:|:------------:|:---------:|:------:|
> |     FT     | 0.8857 |    0.5155    |   0.8014  | 0.3534 |
> |  GeoCon-32 | 0.9557 |    0.5521    |   0.8414  | 0.4695 |
> | GeoCon-256 | 0.9623 |    0.5775    |   0.8552  | 0.4819 |
>
> >Q6. The way you measure CG is very similar to Xu et al. (2022) and Mahon et al. (2024).
>
> A6. We have already added the reference of Mahon et al. (2024) in the updated version. Could you please demonstrate the information of Xu et al. (2022) for us as a reference?
>
> >Q7. This is too strong a statement in my view. On whether there is a connection between disentanglement and CG, there are papers on both sides: those you cite claim there is not. If you are to take a side on that issue, you could measure disentanglement on the embeddings you take from the frozen models in Section 3.2, and see if it correlates with CG and maybe also PS. That would be an interesting addition to your work.
>
> A7. Thanks for your correcting and sorry for my imprecise statement. We have already rectified this fault in the updated version.
>
> Also, thank you for your suggestions regarding the relationship between disentanglement and compositional generalization. This is a very interesting topic. Due to time constraints, we will attempt to enrich this part of the content in the camera-ready version.
>
> >Q8. Does 'accuracy' in Thm. 1 refer to train set accuracy?
>
> A8. Sorry for the misleading. Here "accuracy" refers to test accuracy rather than train accuracy. Theorem 1 aims to illustrate that when the stochastic noise is not taken into consideration, PS = 1, and training is conducted under the CG setting, if the constraint of MinDV regularization is added, as shown in Equation 10, then for the out-of-combination test samples, completely accurate classification can also be achieved. In contrast, a vanilla classifier may fail, referring to Figure 4 (b) for understanding.
>
> >Q9. Other minor points.
>
> A9. We have already rectified these faults by blue text in the updated version.

---

> > ### Comment · Reviewer_9Z46 · 2024-11-21
> >
> > Thanks for your reply.
> >
> > Xu et al. (2022) measures CG by excluding certain combinations from training, then training a simple model to predict the excluded combinations from the encodings after training. See Section 3.2 of that paper.
> >
> > The passage you've updated re DE and CG now is confusing to me. Mahon et al. says there *is* a connection between DE and CG, while Montero et al. says there is not. Maybe you are missing a 'not' somewhere.
> >
> > I'd recommended adding the batch-32 results to the supplementary material.

---

> > > ### Author Response · Authors · 2024-11-21
> > >
> > > Dear Reviewer 9Z46,
> > >
> > > Thank you for your careful reading and timely reply.
> > >
> > > - Thanks for your supplement. We have added the citations of Xu et al. (2022) and Mahon et al. (2024) at Line 140, which measure CG in a similar way to ours.
> > >
> > > - Sorry for omitting "not" and thus making a wrong statement. We have already revised the expression at Line 50 to make it more rigorous. Thank you for your careful reading and correction.
> > >
> > > - In addition, we have also supplemented the latest experimental results in Appendix E for reference.
> > >
> > > We believe that these updates will strengthen our paper and enhance its clarity. We look forward to addressing any further questions you may have, and we would be grateful if you could consider increasing the review score.

---

> > > ### Author Response · Authors · 2024-11-25
> > >
> > > Dear Reviewer 9Z46,
> > >
> > > We appreciate your review of our paper and the valuable feedback you provided. We have carefully studied your suggestions, made modifications, and provided explanations.
> > >
> > > As the rebuttal period is coming to an end soon, we are writing to remind you that we have already provided some feedback regarding your review. We sincerely hope these can address your concerns.
> > >
> > > If you have any other questions or concerns, please contact us at your convenience. We are committed to ensuring that all aspects of the submission are thoroughly addressed.
> > >
> > > Thank you for your time.

---

> > > ### Author Response · Authors · 2024-11-30
> > >
> > > Dear Reviewer 9Z46,
> > >
> > > We appreciate your review of our paper and the valuable feedback you provided. We have carefully studied your suggestions, made modifications, and provided explanations.
> > >
> > > As the rebuttal period is coming to an end soon, we are writing to remind you that we have already provided some feedback regarding your review. We sincerely hope these can address your concerns.
> > >
> > > If you have any other questions or concerns, please contact us at your convenience. We are committed to ensuring that all aspects of the submission are thoroughly addressed.
> > >
> > > Thank you for your time.

---

> > > > ### Comment · Reviewer_9Z46 · 2024-12-01
> > > >
> > > > Thank you for the replies and updates to the paper. These have broadly addressed my concerns and I am increasing my score to reflect this.

---

### Official Review · Reviewer_Lx8X · 2024-11-02

**Soundness:** 2
**Presentation:** 3
**Contribution:** 2
**Rating:** 5
**Confidence:** 4

**Summary:**

In this submission, the authors proposed a neuroscience-inspired method to tackle the problem of generalization of deep neural networks. Specifically, the authors borrow the idea of abstract representation in Ito NeurIPS 2022 to improve the compositional generalization in deep neural networks by applying geometric constraints to the representation structure, including maximizing the parallelism score of representations in the featurizer and minimizing the distance variance between sample points and decision boundary in the classifier. Results are reported on several benchmark dataset, which shows the effectiveness of the proposed method compared with linear probeing and variants of fine-tuning methods.

**Strengths:**

The idea of introducing the concept from neuroscience to the generalization problem of deep neural networks is interesting.

**Weaknesses:**

- The idea is coming from Ito, NeurIPS 2022, which is also based on compositional generalization for neural networks. To me, the main idea is quite similar, except that the authors introduce more regularization terms (also introduced in the Ito paper) in the loss function, which is quite incremental. Can you please elaborate more on how the proposed method advances Ito, NeurIPS 2022, in case I miss something? Moreover, is it possible also to use Ito 2022 as a baseline, where the generalization capability of neural networks has been discussed (see Ito Sec 3.5)?

- While the method's evaluation focuses on class and style, and draws motivation from neuroscience, it's unclear how it advances the state-of-the-art compared to existing statistical approaches to invariant feature learning that achieve similar results.

- While the proposed method is neuroscience-based, the results discussion is only limited to accuracy on domain generalization, which is not satisfactory. How does it benefit other generalization tasks, such as zero-shot? explainable AI?

-  I'd also like to suggest the authors consider the dataset in Ito, NeurIPS 2022, which can better help the reader to have a deeper understanding of the deep relationship between generalization and neural science (which is the main claim of the submission).

**Questions:**

please see above.

---

> ### Author Response · Authors · 2024-11-20
>
> Dear Reviewer Lx8X,
>
> Thank you for taking the time to review. We appreciate that you are interested in our research topic and proposed method. We would like to clarify the following concerns. If you find these responses useful in addressing your questions and concerns, we would be grateful and would consider increasing your score.
>
> >Q1. To me, the main idea is quite similar, except that the authors introduce more regularization terms (also introduced in the Ito paper) in the loss function, which is quite incremental. How does the proposed method advance Ito, NeurIPS 2022?
>
> A1. In fact, Ito et al. (NIPS 2022) do not propose any regularization terms but only verify the relationship between the PS and CG capability. As for the applications in ANNs, they adopt the methods of primitives pre-training and simple task pre-training instead of using geometric regularization constraints. As far as we know, we are the first paper to design PS as a regularizable term that can be optimized. Meanwhile, on this basis, we have also proposed the MinDV regularization term to further enhance the CG capability. Compared with the heuristic methods in Ito et al. (NIPS 2022), the design of our method is inspired by neuroscience, verified through simulation studies and empirical studies, and owned theoretically guaranteed.
>
> >Q2. Is it possible also to use Ito, NeurIPS 2022 as a baseline? I'd also like to suggest the authors consider the dataset in Ito, NeurIPS 2022.
>
> A2. First of all, we need to emphasize that the task setting between Ito et al. (NIPS 2022) and ours is quite different. The concept factors are also prediction targets in our task, implying that we have multiple prediction targets, while Ito et al. (NIPS 2022) regards concept factors as attributes and aims to predict a new target that is not contained in the concept factors, and only has just one prediction target. Therefore, it is difficult to utilize their method for our task or employ our method on their dataset. What's more, their code is not open source and their dataset is huge (over 400GB), which brings some obstacles.
>
> Even so, we try to understand the essence of their method and validate it in our tasks. Specifically, their method doesn't random sample from mixing all the composition data. Instead, they try to learn a single concept first and then extend to some new compositions. The core of their algorithm is to start from simple settings and then gradually expand to complex compositions. Since their method is not open-source, we refer to Appendix 7 and 8 in Ito et al. (NIPS 2022) and implement their algorithm through the following approach: First, we conduct training on two randomly selected groups of concepts (on the four corresponding compositions), and then gradually expand to new compositions one by one until all the compositions in the train set have been trained. The results are demonstrated as follows, where primitives pre-training (PPT) refers to their method. We find that, compared with FT, PPT shows an improvement when FT doesn't work well, but is still inferior to our GeoCon method.
>
> |       Method      |  PACS  |  Office-Home | DomainNet |  NICO  | Shapes3D |
> |:-----------------:|:------:|:------------:|:---------:|:------:|:--------:|
> | FT / from Scratch | 0.8857 |    0.5155    |   0.8014  | 0.3534 |  0.5145  |
> |        PPT        | 0.8838 |    0.5324    |   0.8030  | 0.4012 |  0.6094  |
> |       GeoCon      | 0.9623 |    0.5775    |   0.8552  | 0.4819 |  0.7374  |

---

> ### Author Response · Authors · 2024-11-20
>
> >Q3. How it advances the state-of-the-art compared to existing statistical approaches to invariant feature learning?
>
> A3. Initially, invariant feature learning mostly is researched in domain generalization (DG) and compositional zero-shot learning (CZSL). Compared to algorithms for DG and CZSL, whose goal is to learn invariant features, our framework aims to learn features containing information of all concepts, with strong disentanglement capability at the same time. It means that DG and CZSL hope to learn features only including object-related features, not including style-related features. But our CG aims to learn not only object-related but also style-related features, and maintain the capability to differentiate them.
>
> In addition, our CG has two prediction targets, compared to only one prediction target of DG and CZSL. The difference in training settings causes most of the DG and CZSL algorithms to have difficulties in employing in our CG setting.
>
> Moreover, we have already demonstrated the results of LP-FT and WiSE-FT in our CG task in Table 2. LP-FT and WiSE-FT are the state-of-the-art methods for CLIP fine-tuning, which train on a single in-distribution domain and test on remaining out-of-distribution domains, aiming to learn invariant features. As shown in the following table, our GeoCon method outperforms them.
>
> |  Method |  PACS  |  Office-Home | DomainNet |  NICO  |
> |:-------:|:------:|:------------:|:---------:|:------:|
> |  LP-FT  | 0.9066 |    0.5520    |   0.8021  | 0.3615 |
> | WiSE-FT | 0.9109 |    0.5433    |   0.8233  | 0.4151 |
> |  GeoCon | 0.9623 |    0.5775    |   0.8552  | 0.4819 |
>
> Finally, we also analyze the underlying principle between our MinDV regularization and invariant risk minimization (IRM). As stated in Line 290, both MinDV and IRM are based on the assumption that there exists a substantial disparity between the observed training distribution and the unseen testing distribution. They strive to ensure that the model performs similarly across all visible distributions, rather than significantly outperforming on a specific distribution.
>
> >Q4. How does it benefit other generalization tasks, such as zero-shot? explainable AI?
>
> A4. Referring to the setting of compositional zero-shot learning, in our CG setting, the unseen compositions can be understood as zero-shot data, which is the relationship with zero-shot learning. Regarding explainability, our method attempts to demystify the mechanism of compositional generalization through the geometric structure of brain-inspired representations, and it is theoretically guaranteed. Therefore, to a certain extent, our method also realizes explainable AI from the perspectives of neuroscience, mathematics, and geometry. Meanwhile, our method also has the potential for application in multimodal tasks, you can refer to Answer 5 for Reviewer L2Dd for detailed information.

---

> ### Author Response · Authors · 2024-11-23
>
> Dear Reviewer Lx8X,
>
> We appreciate your review of our paper and the valuable feedback you provided. We have carefully studied your suggestions, made modifications, and provided explanations. We sincerely hope that you will continue to engage in the discussion.
>
> Furthermore, we hope these revisions and clarifications will prompt you to reconsider your evaluation, as these updates directly address your constructive feedback.
>
> If you have any other questions or concerns, please contact us at your convenience. We are committed to ensuring that all aspects of the submission are thoroughly addressed.
>
> Thank you for your time and consideration.

---

> > ### Comment · Reviewer_Lx8X · 2024-11-23
> > **Thansk for the response, but many things still not clear**
> >
> > - I agree with the authors that the proposed method and the one in Ito is different, which is also highlighted in my previouis commnets, but the contribution by only adding regualazation term is marginal to me.
> >
> > - Since the proposed method is based on Ito's method, it is still not clear to me why Ito method can be applied as a baseline. To me, you propose A+B, using A as baseline is straightforward, but please correct me if i got it wrong.
> >
> > - the claim "But our CG aims to learn not only object-related but also style-related features, and maintain the capability to differentiate them." may not hold, many disentanglement based methods are designed based on object-agnostic assumption.
> >
> > To sum up, I keep my score unchanged.

---

> > > ### Author Response · Authors · 2024-11-23
> > >
> > > Dear Reviewer Lx8X,
> > >
> > > >I agree with the authors that the proposed method and the one in Ito is different, which is also highlighted in my previouis commnets, but the contribution by only adding regualazation term is marginal to me.
> > >
> > > We apologize for the misunderstanding maybe caused by the writing, which leads you to think that our work is just an A + B approach. Herein, we would like to reclarify our novelty, especially when compared with those of Ito et al. (NIPS 2022):
> > >
> > > 1. **Motivation**: Our work and Ito et al. (NIPS 2022) are more of a parallel relationship rather than an incremental one, and the motivations of the two are completely different. Specifically, our methods all originate from Bernardi et al. (Cell 2020), where they initially mentioned the parallelism score in neuroscience. Based on their work, the discussion of Ito et al. (NIPS 2022) is still confined to the field of neuroscience, verifying on their own data and proposed a new pre-training task for a simple MLP. In contrast, our main intention is to study whether DNNs exhibit the same PS performance as in neuroscience in visual tasks, especially the relationship between the CG capability and PS of the representations automatically generated by those pre-trained models. Therefore, a large part of the article is devoted to discussing the results of numerous simulation studies and empirical studies.
> > >
> > > 2. **Insights**: Regarding the spontaneously generated capabilities of pre-trained models in terms of PS and CG, we provide many insights. Besides regularization, we also systematically study the relationship between the CG capability and PS of existing pre-trained models, which have never been done before. Based on these empirical studies, we propose numerous insights, including the impacts of different pre-training strategies, different model sizes, and different dataset scales, as well as the relationship between the performance of different concepts. These inspirations should not be ignored. It is precisely due to the problems existing in the current DNNs that we consider how to improve their CG capability through optimization methods.
> > >
> > > 3. **Methods**: Directly applying PS to DNNs doesn't work. Regarding the MinDV regularization, it has nothing to do with Ito et al. (NIPS 2022). It focuses more on the design of the classifier to improve the CG capability after obtaining the existing representations, and the optimization of CG requires its assistance. For the MaxPS regularization, we aim to apply a metric from the field of neuroscience to deep neural networks and design it as an optimizable term. These cross-domain adaptations and the design of the metric as an optimization target are complex. Specifically, we need to give a strict mathematical definition of the parallelism score, and this item can be optimized by the SGD algorithm. We also need to consider how to calculate PS as efficiently as possible within a batch.
> > >
> > > In summary, we have essential differences from Ito et al. (NIPS 2022) in aspects of motivation, insights, methods, and modality. Our work is parallel rather than incremental. We hope these clarifications will prompt you to reconsider your evaluation.
> > >
> > > Bernardi et al. (Cell 2020). The Geometry of Abstraction in the Hippocampus and Prefrontal Cortex.

---

> > > ### Author Response · Authors · 2024-11-23
> > >
> > > >Since the proposed method is based on Ito's method, it is still not clear to me why Ito method can be applied as a baseline. To me, you propose A+B, using A as baseline is straightforward, but please correct me if i got it wrong.
> > >
> > > Once again, we would like to reaffirm that our method is not based on the A + B work of Ito et al. (NIPS 2022). Furthermore, as per your request, we have already taken Ito et al. (NIPS 2022) as a baseline for the experiments. Here, we have already presented the results of comparison experiments with the PPT method of Ito et al. (NIPS 2022), and our method outperforms it significantly.
> > >
> > > |       Method      |    PACS    |  Office-Home |  DomainNet |    NICO    |  Shapes3D  |
> > > |:-----------------:|:----------:|:------------:|:----------:|:----------:|:----------:|
> > > | FT / from Scratch |   0.8857   |    0.5155    |   0.8014   |   0.3534   |   0.5145   |
> > > |        PPT--Ito et al. (NIPS 2022)         |   0.8838   |    0.5324    |   0.8030   |   0.4012   |   0.6094   |
> > > |       GeoCon      | **0.9623** |  **0.5775**  | **0.8552** | **0.4819** | **0.7374** |
> > >
> > > >the claim "But our CG aims to learn not only object-related but also style-related features, and maintain the capability to differentiate them." may not hold, many disentanglement based methods are designed based on object-agnostic assumption.
> > >
> > > Sorry for the imprecise expression. Besides that, we have also compared the results of LP-FT and WiSE-FT, which are methods of invariant feature learning.
> > >
> > > |  Method |    PACS    |  Office-Home |  DomainNet |    NICO    |
> > > |:-------:|:----------:|:------------:|:----------:|:----------:|
> > > |  LP-FT  |   0.9066   |    0.5520    |   0.8021   |   0.3615   |
> > > | WiSE-FT |   0.9109   |    0.5433    |   0.8233   |   0.4151   |
> > > |  GeoCon | **0.9623** |  **0.5775**  | **0.8552** | **0.4819** |
> > >
> > > We look forward to addressing any further questions you may have. If you have any other questions or concerns, please contact us.

---

> > > ### Author Response · Authors · 2024-11-24
> > >
> > > Dear Reviewer Lx8X,
> > >
> > > We appreciate your review of our paper and the valuable feedback you provided. We have carefully studied your suggestions, made modifications, and provided explanations.
> > >
> > > As the rebuttal period is coming to an end soon, we are writing to remind you that we have already provided some feedback regarding your review. We sincerely hope these can address your concerns.
> > >
> > > If you have any other questions or concerns, please contact us at your convenience. We are committed to ensuring that all aspects of the submission are thoroughly addressed.
> > >
> > > Thank you for your time.

---

> > > ### Author Response · Authors · 2024-11-25
> > >
> > > Dear Reviewer Lx8X,
> > >
> > > We appreciate your review of our paper and the valuable feedback you provided. We have carefully studied your suggestions, made modifications, and provided explanations.
> > >
> > > As the rebuttal period is coming to an end soon, we are writing to remind you that we have already provided some feedback regarding your review. We sincerely hope these can address your concerns.
> > >
> > > If you have any other questions or concerns, please contact us at your convenience. We are committed to ensuring that all aspects of the submission are thoroughly addressed.
> > >
> > > Thank you for your time.

---

> > > ### Author Response · Authors · 2024-11-30
> > >
> > > Dear Reviewer Lx8X,
> > >
> > > We appreciate your review of our paper and the valuable feedback you provided. We have carefully studied your suggestions, made modifications, and provided explanations.
> > >
> > > As the rebuttal period is coming to an end soon, we are writing to remind you that we have already provided some feedback regarding your review. We sincerely hope these can address your concerns.
> > >
> > > If you have any other questions or concerns, please contact us at your convenience. We are committed to ensuring that all aspects of the submission are thoroughly addressed.
> > >
> > > Thank you for your time.

---

> > > ### Author Response · Authors · 2024-12-01
> > >
> > > Dear Reviewer Lx8X,
> > >
> > > Thank you once again for reviewing our paper and providing valuable feedback. We have provided explanations and additional experimental results regarding the novelty and the new baseline in our paper. As the rebuttal stage is coming to an end, we sincerely hope that you will continue to engage in further discussions.
> > >
> > > If you have any other questions or concerns, please feel free to contact us at any time. We are committed to ensuring that all aspects of the submission are thoroughly addressed.
> > >
> > > Thanks for your time and consideration.

---

### Official Review · Reviewer_L2Dd · 2024-11-04

**Soundness:** 2
**Presentation:** 3
**Contribution:** 2
**Rating:** 6
**Confidence:** 3

**Summary:**

The paper provides a mathematical formulation of parallelism score (PS) from neuroscience, and how it contributes to out-of-combination compositional generalization (CG) in deep neural networks consisting of a featurizer and classifier for classification tasks. Initial experiments on synthetic datasets reveal a strong positive correlation between PS and CG ability, but also high variance with low CG ability even for high PS. The authors propose minimum distance variance (DV) regularization to reduce such failure cases which encourages similar distance of all samples from the decision boundary. The authors also evaluate multiple pretrained models on real world datasets like PACS, NICO, and Office-Home consisting of class and domain labels, and found that the compositional generalization and parallelism score for each class is strongly correlated. However, there is a negative correlation between compositional generalization for class and domain. Self-supervised DINO and multimodal CLIP models demonstrate good compositional generalization capability for class and domain. The authors finally propose the geometric constraint (GeoCon) method maximizing PS for the featurizer and minimizing DV for the classifier to further improve CG ability across both class and domain.

**Strengths:**

1. The paper is well-written and easy to follow.

2. A formal description of the parallelism score from neuroscience is provided.

3. The authors propose GeoCon method which consists of two regularization techniques, one for distance variance minimization and the other for parallelism score maximization to improve the compositional generalization ability of deep neural networks.

4. The authors demonstrate the effectiveness of their method on multiple real world datasets, and the importance of both regularization techniques through ablation studies.

**Weaknesses:**

1. The GeoCon method seems to require a similar number of classes for each concept.

2. The experiments are limited to datasets with only two concepts.

3. It is not clear how the method would scale to more realistic tasks, like visual question answering.

**Questions:**

1. How does the GeoCon method generalize when there are more than two concepts?

2. Can the authors share more details about the linear probing and finetuning baselines?

3. In Figure 8, shouldn’t the values of the parallelism score and distance variance for the models be similar at epoch=0, which is just after initialization?

4. Is the method only limited to classification problems or more generally applicable to other domains like compositional visual reasoning (e.g. COLA [1] benchmark)?

5. How many samples are required to compute the parallelism score?

[1] - Ray, A., Radenovic, F., Dubey, A., Plummer, B., Krishna, R. and Saenko, K., 2024. Cola: A benchmark for compositional text-to-image retrieval. Advances in Neural Information Processing Systems, 36.

---

> ### Author Response · Authors · 2024-11-20
>
> Dear Reviewer L2Dd,
>
> Thank you for your positive and thoughtful feedback and look into every detail of our work. We appreciate that you found our paper well-written and experiments commendable. We summarize the mentioned concerns and we hope the corresponding comments address your concerns, and we would be grateful if you could consider increasing the review score.
>
> >Q1. The GeoCon method seems to require a similar number of classes for each concept.
>
> A1. Our GeoCon method also works when the number of categories for each concept has a great disparity, for example, in the experiments on the Shapes3D dataset, the object hue has 10 classes, while the shape has 4 classes.
>
> Since a larger number of categories will lead to greater difficulty, leading to the lower classification accuracy of the concepts. What we are concerned about is the accuracy of classifying two concepts correctly at the same time, which is easily affected by the lower accuracy of concept with a larger number of categories. This will cause the model to only focus on learning the tasks related to the concept with a larger number of categories as much as possible, while ignoring the balance between the two concepts. Therefore, for better evaluation, we artificially set the number of categories for each concept to be approximately the same.
>
> >Q2. How does the GeoCon method generalize when there are more than two concepts?
>
> A2. To facilitate comprehension and maintain simplicity, we just discuss the situation when the concept factor is two. Here we will add detailed information when there are more than two concepts:
>
> When the quantity of concept factors is N, there are N targets to be predicted. For the featurizer, they will share a public one as a visual encoder to extract features. For classifiers, we arrange N independent linear functions to predict N concepts. For MinDV regularization, we calculate distance variance within N concepts, indicating that N regularization terms will exist for N concepts. For MaxPS regularization, to compute PS, two groups of concept factors are needed, and we regard other concept factors as noise, like what we do in the Shapes3D dataset. We traverse all possible compositions of any two concepts and sum them up to get the final result.
>
> We test our results when there are three concepts in the Shapes3D dataset: object hue, shape, and wall hue. The separation of train set and test set brings into correspondence with Section 4.1. The results of different methods are shown as follows, which validate our GeoCon method's effectiveness.
>
> |   Method  | Acc-object hue | Acc-wall hue | Acc-shape | Acc-all |
> |:---------:|:--------------:|:------------:|:---------:|:-------:|
> |  baseline |     0.6910     |    0.7531    |   0.5434  |  0.2853 |
> | w/o MaxPS |     0.7018     |    0.7271    |   0.5849  |  0.3145 |
> | w/o MinDV |     0.8187     |    0.8566    |   0.7023  |  0.4839 |
> |   GeoCon  |     0.8405     |    0.8842    |   0.7341  |  0.5241 |
>
> >Q3. Can the authors share more details about the linear probing and finetuning baselines?
>
> A3. For linear probing and fine-tuning, we extend two random-initialized linear functions as classifiers after the featurizer. We set the total epochs as 100, batch size as 256, and utilize the SGD as the optimization method. For the featurizer, the initial learning rate is 1e-3 (only for fine-tuning), and for the classifiers, the initial learning rate is 1e-2. We employ the ReduceLROnPlateau as the learning rate scheduler, with a factor of 0.1 and patience of 5, and the evaluation metric is set as CG-both on the training set. We will release our code after it is accepted.
>
> >Q4. In Figure 8, shouldn’t the values of the parallelism score and distance variance for the models be similar at epoch=0, which is just after initialization?
>
> A4. Sorry for the misleading. You are right and thanks for correcting. In the previous version, we mistakenly recorded the result of the 5th epoch as the first value. Instead, the result of the randomly initialized model should have been used as the first value here. Meanwhile, we also recorded the results every 5 epochs. In the updated version, we have corrected this fault and recorded the results of each epoch within the range of [0, 200].

---

> ### Author Response · Authors · 2024-11-20
>
> >Q5. Is the method only limited to classification problems or more generally applicable to other domains like compositional visual reasoning (e.g. COLA [1] benchmark)?
>
> A5. This is an interesting and open question. In this paper, we mainly discuss the classification task, which aims to reveal the underlying geometric mechanisms of CG, paying more attention to interpretability analysis from a neuroscience perspective rather than applications. As for multimodal tasks, CG also serves as a challenging and significant problem. Our method has the ability to be applied to multimodal tasks, which is also our ongoing work. Due to time constraints, we present a potential solution here. We will try to present some results in the camera-ready version.
>
> For example, Yuksekgonul et al. (ICLR 2023) and Zhang et al. (CVPR 2024) also attempt to resolve CG problems in multimodal tasks. They usually conduct finer-grained alignment by constructing hard negative samples. For example, they set a threshold between hard negative samples and simple negative samples as a hyperparameter to continue with contrastive learning. In fact, MaxPS can also be regarded as a type of contrastive learning, and we can also achieve finer-grained alignment by adding the MaxPS regularization into the contrastive learning loss function. For instance, the transform vector of the positive text sample and the hard negative text sample should be as parallel as possible to the transform vector of the corresponding image samples, and the transform vector from the image feature to the text feature of the positive sample should be as parallel as possible to the transform vector from the image feature to the text feature of the hard negative sample.
>
> In this process, one challenge when adapting our GeoCon method to multimodal tasks is that it is difficult to find a single change concept factor in different settings, referring to the concept transform vector in Equation 4 and 5, especially in image data. Once this problem is be solved, our method can be easily adapted into various multimodal tasks. Tong et al. (CVPR 2024) propose the MMVP benchmark based on the difference between CLIP score and DINO score, which can find image pairs with subtle differences in detail. These image pairs can help us identify a single change concept factor in image data. This is what we are currently exploring, applying our method to more multimodal models.
>
> [1] Yuksekgonul et al. When and Why Vision-Language Models Behave Like Bags-of-Words, and What to Do About It? ICLR 2023.
>
> [2] Zhang et al. Contrasting Intra-Modal and Ranking Cross-Modal Hard Negatives to Enhance Visio-Linguistic Compositional Understanding. CVPR 2024.
>
> [3] Tong et al. Eyes Wide Shut? Exploring the Visual Shortcomings of Multimodal LLMs. CVPR 2024.
>
> >Q6. How many samples are required to compute the parallelism score?
>
> A6. It is a very essential problem. To get an absolutely accurate PS, you need to calculate all samples of the dataset. According to Equation 14, the minimal sample quantity required to calculate PS is 4 with the target concepts of (a,b), (a',b), (a,b'), (a',b').
>
> As stated in Section 3.2, we hope to generate an estimation of PS as accurately as possible. The batch size depends on the specific dataset, including the number of concept factors and the number of target categories. More concept factors and target categories imply a larger batch size. In practice, we set the batch size as 256.
>
> However, when the batch size is limited, as mentioned in Line 364, we can employ an exponential smoothing method to reduce the estimation variance, thus resolving the challenge.
>
> To demonstrate that our method remains effective even with smaller batch size, we conduct experiments with batch size 32. Compared to GeoCon with batch size 256, there was only a negligible drop in performance, while outperforming the FT baseline much.
>
> |   Method   |  PACS  |  Office-Home | DomainNet |  NICO  |
> |:----------:|:------:|:------------:|:---------:|:------:|
> |     FT     | 0.8857 |    0.5155    |   0.8014  | 0.3534 |
> |  GeoCon-32 | 0.9557 |    0.5521    |   0.8414  | 0.4695 |
> | GeoCon-256 | 0.9623 |    0.5775    |   0.8552  | 0.4819 |

---

> ### Author Response · Authors · 2024-11-23
>
> Dear Reviewer L2Dd,
>
> We appreciate your review of our paper and the valuable feedback you provided. We have carefully studied your suggestions, made modifications, and provided explanations. We sincerely hope that you will continue to engage in the discussion.
>
> Furthermore, we hope these revisions and clarifications will prompt you to reconsider your evaluation, as these updates directly address your constructive feedback.
>
> If you have any other questions or concerns, please contact us at your convenience. We are committed to ensuring that all aspects of the submission are thoroughly addressed.
>
> Thank you for your time and consideration.

---

> ### Author Response · Authors · 2024-11-24
>
> Dear Reviewer L2Dd,
>
> We appreciate your review of our paper and the valuable feedback you provided. We have carefully studied your suggestions, made modifications, and provided explanations.
>
> As the rebuttal period is coming to an end soon, we are writing to remind you that we have already provided some feedback regarding your review. We sincerely hope these can address your concerns.
>
> If you have any other questions or concerns, please contact us at your convenience. We are committed to ensuring that all aspects of the submission are thoroughly addressed.
>
> Thank you for your time.

---

### Note · Authors · 2025-02-01

**Comment:**

We would like to express our gratitude to all reviewers, AC, and PC for their efforts. Particularly, Reviewer 9Z46 found our experimental results to be convincing, and Reviewer L2Dd considered our method to be effective and significant.

However, we believe that Reviewer Lx8X still misunderstood the novelty of our work and neglected the new experimental results presented during the rebuttal period. Meanwhile, we also believe that timely responses and discussions during the rebuttal period are necessary, rather than having long periods of no response like Reviewer Lx8X, which might have influenced the final decision of the AC.

Overall, we will also take into account the valuable comments from the reviewers and present a more refined version in the future.

**Withdrawal Confirmation:**

I have read and agree with the venue's withdrawal policy on behalf of myself and my co-authors.